# Improved Confidence Regions and Optimal Algorithms for Online and Offline Linear MNL Bandits

**Yuxuan Han**
New York University
yh6061@stern.nyu.edu

**Jose Blanchet**
Stanford University
jose.blanchet@stanford.edu

**Zhengyuan Zhou**
New York University
zzhou@stern.nyu.edu

## Abstract

In this work, we consider the data-driven assortment optimization problem under the linear multinomial logit (MNL) choice model. We first establish an improved confidence region for the maximum-likelihood-estimator (MLE) of the $d$-dimensional linear MNL likelihood function that removes the explicit dependency on a problem-dependent parameter $\kappa^{-1}$ in previous result [42], which scales exponentially with the radius of the parameter set. Building on the confidence region result, we investigate the data-driven assortment optimization problem in both offline and online settings. In the offline setting, the previously best-known result scales as $\widetilde{O}\left(\sqrt{\frac{d}{\kappa n_{S^\star}}}\right)$, where $n_{S^\star}$ denote the number of times that optimal assortment $S^\star$ is observed [26]. We propose a new pessimistic-based algorithm that, under a burn-in condition, removes the dependency on $d, \kappa^{-1}$ in the leading order bound and works under a more relaxed coverage condition, without requiring the exact observation of $S^\star$. In the online setting, we propose the first algorithm to achieve $\widetilde{O}(\sqrt{dT})$ regret without a multiplicative dependency on $\kappa^{-1}$. In both settings, our results nearly achieve the corresponding lower bound when reduced to the canonical $N$-item MNL problem, demonstrating their optimality.

## 1 Introduction

In modern data-driven decision-making problems, a key challenge for sellers, often managing a large inventory, is determining a subset of products, also referred to as an *assortment*, to display to customers. For instance, when customers search for "laptops" on an e-commerce platform, the system must select an assortment from thousands of options to present. Due to space constraints or cognitive overload, only a limited number of items—at most $K$—can be displayed at a time. This motivates the study of *assortment optimization* with *cardinality constraints*, where the seller aims to identify the optimal assortment of $K$ items to display in order to maximize revenue.

The revenue of an assortment $S$ depends on both the revenue of each item $i \in S$ and the final choice made by the customer after observing $S$. While the revenue of individual items is often known to the seller, the customer's choice behavior is unknown and must be estimated from data. A large body of work has focused on modeling customer choice behavior in the context of assortment optimization [40, 52, 41, 7, 22, 25, 6, 12]. Among these models, the multinomial logit (MNL) model and its linearly parametrized variant stands out as a widely used approach and serves as a foundation for understanding more complex models due to its clear mathematical structure, computational simplicity, and ease of calibration [11].

In the linear MNL choice model with $N$ items, each item $i \in [N]$ is associated with an $d$-dimensional feature $x_i$, and its utility, presenting its attraction to customers, is given by $v_i = x_i^\top \theta^\star$ for some

39th Conference on Neural Information Processing Systems (NeurIPS 2025).

| Offline Learning | | | |
|---|---|---|---|
| | Leading-Order Upper Bound | Lower Bound | Remark |
| [26] | $\widetilde{O}(\sqrt{d/(\kappa' n_{S^\star})})^\dagger$ | – | $n_{S^\star} = \sum_{m=1}^n \mathbf{1}\{S_m = S^\star\}$ |
| [29] | $\widetilde{O}(K/\sqrt{n^\star})$ | $\Omega(K/\sqrt{n^\star})$ | $N$-item setting, $n^\star = \min_{i\in S^\star}\sum_{m=1}^n \mathbf{1}\{i \in S^\star\}$ |
| Our work | $\widetilde{O}\big(\sqrt{\sum_{i\in S^\star} p_i(S^\star)p_0(S^\star)\|x_i\|^2_{H_D^{-1}(\theta^\star)}}\big)$ $\widetilde{O}\big(\sqrt{d \cdot \sum_{i\in S^\star} p_i(S^\star)p_0(S^\star)\|x_i\|^2_{H_D^{-1}(\theta^\star)}}\big)$ | $\Omega\big(\sqrt{\sum_{i\in S^\star} p_i(S^\star)p_0(S^\star)\|x_i\|^2_{H_D^{-1}(\theta^\star)}}\big)$ | Burn-in condition $\max_{j\in \cup_{m=1}^n S_n} \|x_j\|_{H_D^{-1}(\theta^\star)} \lesssim \frac{1}{\sqrt{d}}$ No burn-in condition |
| Online Learning | | | |
| | Leading-Order Upper Bound | Lower Bound | Remark |
| [5] | $\widetilde{O}(\sqrt{NT})$ | $\Omega(\sqrt{NT/K})$ | $N$-item setting |
| [17] | – | $\Omega(\sqrt{NT})$ | $N$-item setting |
| [18] | $\widetilde{O}(d\sqrt{T})$ | $\Omega(d\sqrt{T}/K)$ | Adversarial context with initial exploration |
| [42] | $\widetilde{O}(\kappa^{-1}\sqrt{dT\log N})$ | – | Adversarial context with initial exploration |
| [44] | $\widetilde{O}(Kd\sqrt{T})$ | – | Adversarial context with uniform reward |
| [35] | $\widetilde{O}(d\sqrt{T})$ | $\Omega(d\sqrt{T})$ | Adversarial context |
| Our work | $\widetilde{O}\left(\sqrt{dT\log N}\right)$ | $\Omega(\sqrt{dT})^\ddagger$ | Fixed design Adversarial context with initial exploration* |

Table 1: Comparison of offline and online assortment optimization results, where $p_i(S)$ denote the choice probability of item $i$ under assortment $S$; only leading-order terms are presented for notational simplicity.

$\dagger$ The $\kappa'$ notation in [26] defined in a different way as those in other works, but the still suffers from the exponential dependency on $\|\theta^\star\|$ in the worst case.

$\ddagger$ When consider the canonical $N$-item setting with $x_i = e_i, d = N$, the $\Omega(\sqrt{NT})$ result in [17] implies this result.

* We leave the related algorithm design and proof to Appendix E.4 due to space limitation.

underlying $\theta^\star \in \mathbb{R}^d$. After given an $S$, the choice of the customer then follows the standard multinomial distribution as described as in (1). While data-driven assortment optimization with the linear MNL model has been extensively studied [14, 46, 50, 4, 5, 19, 48], most of the existing literature focuses on the online learning setting. The current best-known regret bounds are given as $\widetilde{O}(d\sqrt{T} \wedge \kappa^{-1}\sqrt{dT \log N})$. When specialized to the canonical $N$-item setting ($d = N$, $x_i = e_i$ being the canonical basis), the regret either exhibits an additional $\sqrt{d}$ dependence or incurs a multiplicative dependence on $\kappa^{-1}$, a problem-dependent quantity defined in (2), which may scales exponentially with $\|\theta^\star\|$. Besides the online setting, the only known works in the offline learning setting [26, 29] either focus on the canonical $N$-item setting or rely on restrictive assumptions about the coverage of the data.

Motivated by these gaps, we propose new algorithms in this work that achieve improved online regret and offline sample complexity guarantees. Our methods build on a sharper analysis of the linear MNL likelihood function, which serves as the foundation for most existing approaches under the linear MNL model. We compare our results with previous works in Table 1 and summarize our contributions as follows:

## 1.1 Our Contributions

**Sharper Confidence Region result for Maximum Likelihood Estimator.** Our first result is an improved confidence region for the linear MNL maximum likelihood estimator (MLE), which incorporates variance information and avoids explicit dependency on $\kappa$. More precisely, given the conditional independence of the observed choice dataset $D := \{i_k, S_k\}_{k=1}^n$, we show that the corresponding maximizer $\widehat{\theta}$ of the linear MNL likelihood function, under a burn-in condition, satisfies that with high probability, $|x^\top(\widehat{\theta} - \theta^\star)| = \widetilde{O}(\|x\|_{H_D^{-1}(\theta^\star)})$, $\forall \|x\| \leq 1$, with $H_D$ the Hessian matrix of the log-likelihood function given $D$. Our result provides an non-asymptotic variant of the large-sample asymptotic $x^\top(\widehat{\theta} - \theta^\star) \implies \mathcal{N}(0, \|x\|_{H_D^{-1}(\theta^\star)})$ which holds for general $M$-estimators under certain conditions [37], and improves the best previous non-asymptotic result under the same assumptions, stated as $|x^\top(\widehat{\theta} - \theta^\star)| = \widetilde{O}(\kappa^{-1}\|x\|_{V^{-1}})$ [42], since it always holds that $H_D(\theta^\star) \succeq \kappa V$. Our improvement stems from a novel variance-aware analysis combined with a careful exploration of the self-concordant-like properties of the MNL likelihood function, which is inspired by recent developments for MNL likelihood with adaptively collected data [44, 3, 35].

**Offline Assortment Optimization with Item-wise Coverage.** Based on the sharp confidence region result, we then consider the offline assortment optimization problem, where the seller can access to a dataset $D = \{i_k, S_k\}_{k=1}^n$ and aim to find the assortment that maximize the revenue. In this setting, we provide an pessimistic-based algorithm and show that under a basic coverage number of each items, it can achieve the sub-optimality gap that scales with $\widetilde{O}(\sum_{i \in S^\star} p_i(S^\star) p_0(S^\star) \|x_i\|_{H_D^{-1}(\theta^\star)})$ in the leading-order term. Notably, we can show that $p_i(S^\star) p_0(S^\star) \|x_i\|_{H_D^{-1}(\theta^\star)} \gtrsim n_i^{-1/2}$ for $n_i := \sum_{k=1}^n \mathbf{1}\{i \in S_k\}$, thus it suffices for each $i \in S^\star$ to be covered by $S_k$ sufficiently many times in order to ensure that $\|x_i\|_{H_D^{-1}(\theta^\star)}$ becomes small. In contrast, the best-known result for linear MNL model prior to ours, presented in [26], scales as $\widetilde{O}\left(\sqrt{\frac{d}{\kappa n_{S^\star}}}\right)$, where $n_{S^\star} := \sum_{k=1}^n \mathbf{1}\{S^\star = S_k\}$. This result has an additional $d$-dependency and a multiplicative $\kappa^{-1}$-dependency compared to ours. More importantly, their approach requires the optimal assortment $S^\star$ to be exactly observed sufficiently many times, which imposes a restrictive coverage requirement on $D$. Finally, we show that, when reduced to the canonical $N$-item setting with uniform item-wise rewards, our result matches the $\Omega\left(\max_i \sqrt{K/n_i}\right)$ lower bound recently developed in [29]. This demonstrates that the proposed item-wise coverage measure, $p_i(S^\star) p_0(S^\star) \|x_i\|_{H_D^{-1}(\theta^\star)}$, is an appropriate metric for sample complexity in the offline setting.

**Improved Regret for Online Assortment Optimization.** In the online assortment optimization setting, where the seller starts without prior knowledge but can interact with arriving customers over $T$ rounds, we design an algorithm based on the SupCB framework [8] that achieves a regret of $\widetilde{O}(\sqrt{dT \log N} + \kappa^{-1} d)$. This result improves upon the previous regret bound of $\widetilde{O}(\kappa^{-1}\sqrt{dT \log N})$ in [42] by reducing the dependency on $\kappa^{-1}$. Our result also improves the $\widetilde{O}(d\sqrt{T} + \kappa^{-1} d)$ result in [44, 35] on the dependency of $d$ when $N = o(2^d)$. Especially, our result is nearly optimal in the sense that it nearly matches the $\Omega(\sqrt{dT})$ lower bound in [17] when reduced to the canonical $N$-item setting .

## 1.2 Related Works

**Data-Driven Assortment Optimization.** The online assortment optimization problem under the MNL choice model has been extensively studied in the literature [14, 46, 50, 4, 5, 19, 48]. Among these works, [4, 5] and [17] were the first to close the $\widetilde{\Theta}(\sqrt{NT})$ minimax optimal regret for the canonical $N$-item setting. In the linear MNL setting, [18, 44] proposed algorithms achieving a regret of $d\sqrt{T} + \text{Poly}(\kappa^{-1}, d)$, but these algorithms are computationally intractable. While [42] proposed computationally tractable algorithms for the same setting with a regret of $\widetilde{O}(\kappa^{-1} d\sqrt{T} \wedge \kappa^{-1}\sqrt{dT \log N})$, their results depend on $\kappa^{-1}$ in a multiplicative manner. The only known computationally tractable algorithm achieving $\widetilde{O}(d\sqrt{T})$ regret with additive dependency on $\kappa^{-1}$ is that of [35][1]. Our result contributes to this direction by improving the best-known regret bound with additive $\kappa^{-1}$ dependency from $\widetilde{O}(d\sqrt{T})$ to $\widetilde{O}(d\sqrt{T} \wedge \sqrt{dT \log N})$. For the offline assortment optimization problem, the only known works are [26] and [29]. [26] were the first to design a pessimistic-based algorithm for the linear MNL setting. Their algorithm is based on the assortment-wise pessimistic principle, and its performance bound scales to $n_{S^\star}^{-1/2}$, with $n_{S^\star}$ the number of times the optimal assortment $S^\star$ is exactly observed in the dataset. On the other hand, [29] studied the canonical $N$-item setting using an item-wise pessimistic principle. They showed that the minimax rate in this setting is $\frac{K}{\sqrt{\min_{i \in S^\star} n_i}}$, where $n_i$ is the covering time of item $i$ by the observed assortments, thus relaxing the requirement in [26]. The algorithm design in our work can be seen as a generalization of the item-wise pessimistic principle in [29] to the linear setting, with a new concept of item-wise covering introduced. Finally, beyond the MNL setting, learning problems involving additional constraints [20, 10, 15] or other choice models [43, 16, 39, 56, 55] have also been explored.

**Offline Learning via Pessimistic Principle.** Our design of offline algorithms follows the same spirit as the pessimistic (conservative) methods [54, 33] in offline bandit and reinforcement learning. The sample efficiency of pessimistic algorithms under partial coverage in the offline setting has been

---

[1][3] also proposed a computationally tractable algorithm with a similar regret; but their proof contains a technical error, as discussed in Appendix L of [35].

demonstrated in a series of studies [30, 45, 57]. Unfortunately, the setting considered in these works is not directly applicable to the assortment problem, due to differences in the feedback structure.

**Online Learning with Bandit Feedback.** The algorithm design in most online MNL works draws from the optimistic principle, a concept extensively explored in online bandit learning [8, 23, 1, 51, 34]. In particular, the logistic bandit problem can be viewed as a special case of the online assortment problem with $K = 1$, which faces a similar challenge of eliminating the multiplicative dependency on a problem dependent parameter similar to $\kappa^{-1}$, motivating a number of studies [2, 27, 28, 31]. In particular, the confidence region result in our work can be regarded as a generalization of [31] to the MNL setting and our improvement over [42] parallels that of [31] for [38].

## 2 Preliminary

**Revenue Maximization under the Linear MNL Model.** We study the assortment optimization problem, which models the interaction between a *seller* and a *customer*. Let $\{1, \ldots, N\}$ denote the set of $N$ available products/items. An assortment $S \subseteq \{1, \ldots, N\}$ represents the subset of products that the seller offers to the customer. When presented with assortment $S$, the customer chooses a product from the choice set $S_+ = \{0\} \cup S$, where $\{0\}$ represents the no-purchase option. In the $N$-item MNL model, each item has an attraction value $v_i \geq 0$. When a customer encounters assortment $S$, the probability that he/she will choose product $i \in S_+$ is given by

$$p_i(S|\boldsymbol{v}) := \frac{v_i}{1 + \sum_{j \in S} v_j}. \tag{1}$$

Each item $i$ generates a revenue $r_i$ when purchased, while the no-purchase option generates no revenue: $r_0 = 0$. The seller's goal is to maximize the expected revenue from the selected assortment, defined as

$$R(S|\boldsymbol{v}) := \sum_{i \in S} r_i p_i(S|\boldsymbol{v}) = \frac{\sum_{i \in S} r_i v_i}{1 + \sum_{j \in S} v_j}.$$

In the $d$-dimensional linear MNL model with *fixed design*, each item $i \in [N]$ is further associated with a vector $x_i \in \mathbb{R}^d$, and there exists a underlying parameter $\theta^\star \in \mathbb{R}^d$ so that $v_i = \exp(x_i^\top \theta^\star)$. With $\boldsymbol{X} := (x_1, \ldots, x_N) \in \mathbb{R}^{N \times d}$, we also abuse the notation $p_i(S|\theta^\star) := p_i(S|\exp(\boldsymbol{X}^\top \theta^\star))$ and $R(S|\theta^\star) := R(S|\exp(\boldsymbol{X}^\top \theta^\star))$ to denote their dependence on $\theta^\star$ when there is no ambiguity.

Following standard research conventions, we consider the assortments $S$ where $|S| \leq K$. In the following context, we denote $\mathcal{S}_K$ the set consist of all $K$-sized assortments and $S^\star = \arg\max_{S \in \mathcal{S}_K} R(S|\theta^\star)$ the optimal assortment.

Throughout the paper, we assume that the attraction values $v_i$ and the underlying parameter norm $\|\theta^\star\|$ are bounded by constants $V$ and $W$, respectively. We also introduce the problem parameter

$$\kappa := \min_{S \in \mathcal{S}_K} \min_{i \in S} p_i(S|\theta^\star) p_0(S|\theta^\star), \tag{2}$$

it can be seen that $\kappa^{-1}$ can scale with $\exp(W)$ even when $V$ is small. It is worth noting that a series of previous works on MNL bandits [18, 44, 3, 35] focus on improving the dependency on $\kappa^{-1}$ in the sample complexity.

**Offline Assortment Optimization.** In the offline assortment optimization setting, the seller does not know the underlying parameter $\theta^\star$ but can access to a pre-collected dataset $\{i_t, S_t\}_{t=1}^n$ consisting of the choice-assortment pairs, where for each given $S_j$, the corresponding $i_j$ is sampled independently from the linear MNL choice model with parameter $\theta^\star$. The seller's goal is to approximate the optimal assortment based on those observed data, the learning objective is the sub-optimality gap, defined as

$$\text{SubOpt}(S) := R(S^\star|\theta^\star) - R(S|\theta^\star),$$

which measures the gap between the revenue of assortment $S$ and the best-possible revenue.

**Online Assortment Optimization.** In the online learning setting, the seller can interact with the coming customers for $T$ rounds. At each time step $t$, the seller provides an assortment $S_t \in \mathcal{S}_K$ to the customer and then receives a feedback $i_t$ drawn according to the distribution specified in (1). The goal of the seller is to design a policy $\pi = (\pi_1, \ldots, \pi_T)$, where each $\pi_t$ adaptively selects the

assortment $S_t$ based on the historical observations, to minimize the cumulative regret over $T$, defined as

$$\mathrm{Reg}(T) = \mathbb{E}[\sum_{t=1}^{T} \mathrm{SubOpt}(S_t)],$$

which measures the total sub-optimality of the policy $\pi$ over $T$ rounds.

**Notations** Through out the paper, for any real numbers $a, b$ we use the notion $a \vee b$ to denote $\max\{a, b\}$, and we use the notation $a \lesssim b$ if there exists some absolute constant $c > 0$ so that $a < cb$. For matrices and vectors, given any vector $x \in \mathbb{R}^d$ and $A \in \mathbb{R}^{d \times d}$, we denote $\|\cdot\|$ the $\ell_2$-norm, and $\|x\|_A := \sqrt{x^\top A x}$, we also denote $A^\dagger$ the pseudo inverse of $A$.

## 3 Improved Confidence Region for the Linear MNL MLE

In this section, we present our main result on the improved confidence region for the linear MNL MLE. While we focus on the fixed design setting in the main text, where each item's feature $x_i$ remains unchanged throughout $t$, we allow the observed feature $x_{ti}$ to vary across rounds in the dataset to ensure generality in this section's result, and we denote $\boldsymbol{X}_t := \{x_{ti}\}_{i \in [N]}$ throughout this section. Given any dataset $D := \{i_t, \boldsymbol{X}_t, S_t\}_{t=1}^n$, the linear MNL log-likelihood function is defined as

$$\ell_D(\theta) := \sum_{t=1}^{n} \sum_{j \in (S_t)_+} y_{tj} \log p_j(S_t | \exp(\boldsymbol{X}_t^\top \theta)),$$

where $y_{tj} = \mathbf{1}\{i_t = j\}$. In the following context, for any $\lambda \geq 0$ we denote $\widehat{\theta}_D^\lambda$ the $\lambda$-regularized MLE, i.e.,

$$\widehat{\theta}_D^\lambda = \mathrm{argmax}_{\theta \in \mathbb{R}^d} \ell_D(\theta) - \frac{\lambda}{2} \|\theta\|_2^2.$$

Our first result establishes a confidence interval for $x^\top \widehat{\theta}_D^\lambda$ with any $\|x\| \leq 1$ given the following *conditional independence assumption:*

**Assumption 1.** *Condition on $\{(\boldsymbol{X}_t, S_t)\}_{t=1}^n$, the observed choices $\{i_t\}_{t=1}^n$ are mutually independent.*

**Theorem 1.** *Given $D := \{i_t, \boldsymbol{X}_t, S_t\}_{t=1}^n$ and $\widehat{\theta}_D^\lambda$ the $\lambda$-regularized MLE under $D$. Suppose Assumption 1, then for any $x \in \mathbb{R}^d$ with $\|x\| \leq 1$, if we denote $H_D(\theta)$ the Hessian matrix of $\ell_D$ at $\theta$, $H_D^\lambda(\theta) := H_D(\theta) + \lambda I$, and $N_{\text{eff}}$ as the total number of distinct vectors appeared in $\{x_{kj}\}_{j \in S_k, k \leq t}$, we have condition on*

$$64 \max_{k \leq t, j \in S_k} \|x_{k,j}\|_{H_D^\lambda(\theta^\star)^{-1}} \leq \frac{1}{\sqrt{d \log(N_{\text{eff}}/\delta)}} \wedge \frac{1}{\sqrt{\lambda} W},$$

*it holds that with probability at least $1 - \delta$*
*i)*

$$\frac{1}{2} H_D^\lambda(\theta^\star) \preceq H_D^\lambda(\widehat{\theta}_D^\lambda) \preceq 2 H_D^\lambda(\theta^\star).$$

*ii)*

$$|x^\top(\widehat{\theta}_D^\lambda - \theta^\star)| \leq \|x\|_{H_D^\lambda(\theta^\star)^{-1}} \left(36 \sqrt{\log(\frac{N_{\text{eff}}}{\delta})} + 64 \sqrt{\lambda} W \right).$$

Technically, our proof of Theorem 1, detailed in Appendix B, builds on a extended framework for proving Theorem 1 of [31]. But several new ideas that rely on the geometry and self-concordant structure of the $K$-MNL loss are introduced. These structural properties have recently attracted much attention in the theoretical study of MNL bandits and are essential to our argument. The main technical contribution appears in Lemma 6, where we use the curvature of the MNL loss to design a confidence region that does not depend explicitly on $K$ or $\kappa$. This result improves upon [42], which is the only previous work giving $d$-free confidence bounds under the independence condition, and parallels the refinement technique in [36], which is developed for obtaining sharp variance-dependent bounds.

**Comparison to Previous Dimension-Free Result.** Prior to our result, the only confidence interval for the linear MNL MLE achieving a $\sqrt{d}$-free rate is stated in [42]. Their confidence region result scales as $\widetilde{O}(\kappa^{-1}\|x\|_{V_t^{-1}})$ under the burn-in condition

$$\lambda_{\min}(V_t) \geq \kappa^{-4}d^2, \quad V_t = \sum_{k \leq t}\sum_{j \in S_k} x_{k,j}x_{k,j}^{\top}.$$

By the relation $H_t(\theta^{\star}) \succeq \kappa V_t$ and

$$\max_{k \leq t, j \in S_k} \|x_{k,j}\|^2_{H_t^{-1}(\theta^{\star})} \leq \left(\kappa\lambda_{\min}(V_t)\right)^{-1}$$

our result provides an improvement over [42] in both the confidence interval and the burn-in condition.

**Comparison to other Confidence Bound Results.** Besides [42], a line recent works have focused on deriving confidence regions for the MLE that are independent of $\kappa$ [44, 3, 35]. It is worth mentioning that results in [44, 3, 35] no longer require the burn-in condition or the conditional independence assumption, with a price of introducing an additional factor of $\sqrt{d}$ in the resulting confidence interval. The requirement of a burn-in condition to achieve sharp dependency on $d$ first appears in the logistic linear bandit literature, where such a condition arises as [31, 38] refine the results of [27, 2, 28] by improving the dependency on a $\sqrt{d}$ factor. This corresponds to a special case of our setting with $K = 1$. In Appendix F.1, we present several comparison experiments to those burn-in time free bounds to discuss the sensitivity to burn-in condition of Theorem 1.

# 4 Offline Assortment Optimization with Linear MNL Choices

In this section, based on the confidence interval result in Theorem 1, we present the algorithm for offline assortment optimization problem and the corresponding theoretical guarantee.

Throughout this section, we assume W.L.O.G. that $H_D(\theta^{\star})$ is invertible to maintain notational clarity; otherwise, we consider the $\lambda$-regularized MLE and its corresponding confidence region for $\lambda$ very close to 0, which does not affect the theoretical results.

## 4.1 The LCB-LinearMNL Algorithm

For the offline assortment optimization problem, we propose the LCB-LinearMNL algorithm, as detailed in Algorithm 1. The design of Algorithm 1 is inspired by the lower-confidence-bound technique, also known as the pessimistic principle, which has been shown to be theoretically effective in the offline policy learning literature [30, 45, 57]. The LCB-LinearMNL algorithm consists of two steps, the *pessimistic estimation step* and the *revenue maximization step*.

---

**Algorithm 1** LCB-LinearMNL

1: **Input:** Dataset $D = \{(i_j, S_j)\}_{j=1}^n$, feature set $\mathcal{X}$
2: Compute the MLE $\widehat{\theta} = \text{argmax}_{\theta}\ell_n(\theta)$
3: Compute the pessimistic value $v_i^{\text{LCB}}$ as in (3) for each $i \in [N]$.
4: Select the pessimistic assortment: $S^{\text{LCB}} = \text{argmax}_{S \in \mathcal{S}_K} R(S|\boldsymbol{v}^{\text{LCB}})$.
5: **Return:** $S^{\text{LCB}}$

---

**Pessimistic Estimation** In the pessimistic estimation step, with the MLE $\widehat{\theta}_D$, the algorithm first compute

$$u_i^{\text{LCB}} := x_i^{\top}\widehat{\theta}_D - 72\|x_i\|_{H_D^{-1}(\widehat{\theta})}\sqrt{\log(N_{\text{eff}}/\delta)}$$

then takes the pessimistic value estimation for each item $i \in [N]$ via

$$v_i^{\text{LCB}} = \begin{cases} \exp(u_i^{\text{LCB}}) & \text{if } i \in \cup_{j=1}^n S_j, \\ 0 & \text{otherwise} \end{cases}. \tag{3}$$

The confidence region result provided in Theorem 1 ensures the pessimistic property holds for $v_i^{\text{LCB}}$ i.e. $v_i^{\text{LCB}} \leq v_i := e^{x_i^{\top}\theta^{\star}}$ for all $i \in [N]$ with high probability.

**Revenue Maximization** After computing the pessimistic values for each item, the algorithm proceeds to the revenue maximization step, where it selects the assortment that maximizes revenue under $v_i^{\mathrm{LCB}}$. This step can be efficiently solved using several well-studied polynomial-time algorithms [46, 24, 9].

## 4.2 Sub-optimality Gap Guarantee

Now we present the sub-optimality gap guarantee of Algorithm 1.

**Theorem 2.** *Suppose the burn-in condition* $64 \max_{j \in S_k, k \in [n]} \|x_j\|_{H_D^{-1}(\theta^\star)} \leq \left( (d \log(N_{\mathit{eff}}/\delta)) \right)^{-1/2}$ *holds, then with probability at least* $1 - \delta$, *we have*

$$\mathrm{SubOpt}(S^{\mathrm{LCB}}) \leq 36 \sqrt{\sum_{j \in S^\star} p_j(S^\star|\theta^\star) p_0(S^\star|\theta^\star) \|x_j\|_{H_D^{-1}(\theta^\star)}^2} + 36 \max_{j \in S^\star} \|x_j\|_{H_D^{-1}(\theta^\star)}^2.$$

*up to logarithmic factors.*

Theorem 2 shows that the leading-order complexity of the sub-optimality gap depends on the summation of $p_j(S^\star|\theta^\star)p_0(S^\star|\theta^\star)\|x_j\|_{H_D^{-1}(\theta^\star)}$ over $j \in S^\star$, which measures how well the items in the optimal assortment are covered by the observed assortment data. Here we provide a upper bound of the this quantity with respect to item-wise coverage numbers to make the Theorem 2 more transparent:

**Proposition 1.** *Denote* $n_i := \sum_{k \in [n]} \mathbf{1}\{i \in S_k\}$ *and* $n^\star := \min_{i \in S^\star} n_i$, *then it holds that*

$$\sqrt{\sum_{j \in S^\star} p_j(S^\star|\theta^\star) p_0(S^\star|\theta^\star) \|x_j\|_{H_D^{-1}(\theta^\star)}^2} \lesssim \max_m \frac{1 + \sum_{k \in S_m} v_k}{\sqrt{(1 + \sum_{k \in S^\star} v_k) n^\star}}$$

Based on Proposition 1, we can further obtain the following sub-optimality guarantee of Algorithm 1.

**Corollary 1.** *Denote* $n_i := \sum_{k \in [n]} \mathbf{1}\{i \in S_k\}$ *and* $n^\star := \min_{i \in S^\star} n_i$, *then it holds that under the same condition on Theorem 2, we have with probability at least* $1 - \delta$,

$$\mathrm{SubOpt}(S^{\mathrm{LCB}}) \lesssim \max_m \frac{1 + \sum_{k \in S_m} v_k}{\sqrt{(1 + \sum_{k \in S^\star} v_k) n^\star}} + \frac{K}{\kappa n^\star}.$$

*up to logarithmic factors*

**Results under Fixed Data Collecting Policy.** In the setting that $\{S_t\}_{t=1}^n$ is sampled i.i.d. from some fixed exploration policy $\pi^{\mathrm{off}}$, the burn-in condition in Theorem 2 always holds as $n \to +\infty$. Then Theorem 2 implies the following large-sample asymptotic

$$\mathrm{SubOpt}(S^{\mathrm{LCB}}) = \widetilde{O}\left( \frac{1}{\sqrt{n \sum_{j \in S^\star} p_j(S^\star|\theta^\star) p_0(S^\star|\theta^\star) \|x_j\|_{H_{\mathrm{off}}^{-1}}^2}} \right)$$

with $H_{\mathrm{off}} := \mathbb{E}_{S \sim \pi_{\mathrm{off}}} \left[ \sum_{i \in S} p_0(S|\theta^\star) p_i(S|\theta^\star) x_i x_i^\top \right]$ as $n \to +\infty$.

**Comparison to Previous Results.** Prior to our work, the sample complexity of offline assortment optimization under the linear MNL model was largely unexplored. The only known result, from [26], considers a fixed policy $S_t \sim \pi_{\mathrm{off}}$ and yields a complexity of $\widetilde{O}\left( \sqrt{\frac{d/K}{\kappa n \pi_{\mathrm{off}}(S^\star)}} \right)$, which depends on how often the optimal assortment $S^\star$ is sampled. In contrast, our result avoids multiplicative dependence on $d$ and $\kappa^{-1}$ in the leading term and does not require full coverage of $S^\star$. Instead, it suffices for $S^\star$ to be explored through assortments with non-orthogonal feature overlap.

**Result in the Canonical $N$-item setting and the Optimality.** In the *canonical $N$-item MNL setting*, a special case of the linear MNL setting with $d = N$ and $x_j = e_j$, a very recent work [29] shows a similar result to Corollary 1 where the coverage condition for each item is sufficient for learning of the optimal assortment. Their algorithm design shares the same spirit as ours in employing an item-wise pessimistic principle. However, their algorithm and analysis rely heavily on the rank-breaking technique [48, 49, 32] , which is challenging to extend to the general linear MNL setting. More specifically, they propose a $\widetilde{\Theta}(K/\sqrt{n^\star})$ complexity result and a $\widetilde{\Theta}(\sqrt{K/n^\star})$ complexity result in the

uniform reward setting ($r_i \equiv 1$). In particular, Proposition 1 implies the a $\widetilde{O}(K^{3/2}/\sqrt{n^\star})$ upper bound in general setting and an improved $O(\sqrt{K/n^\star})$ bound in the uniform reward setting, since $S^\star$ consists of the top-$K$ items by value. The uniform reward setting result, together with lower bound established in [29], suggests a corresponding lower bound of $\Omega\left(\sqrt{\sum_{j \in S^\star} p_j(S^\star|\theta^\star)p_0(S^\star|\theta^\star)\|x_j\|^2_{H_D^{-1}(\theta^\star)}}\right)$ for the offline linear MNL bandits for uniform reward setting.

**Eliminating the Burn-in Condition.** The burn-in condition in Theorem 2 stems from the confidence region requirement in Theorem 1. The item-wise pessimistic estimation in Algorithm 1 is flexible and can incorporate alternative confidence bounds by replacing (3). In particular, using confidence regions for adaptively collected data (e.g., [42, 44]) yields a burn-in-free guarantee at the cost of an additional $\sqrt{d}$ factor. We state the main result below and defer the proof to Appendix D.2.

**Proposition 2.** *There exists a plug-in confidence region result so that when replacing the pessimistic estimation step in* (3) *by this result, the output Algorithm 1 satisfies*

$$\mathrm{SubOpt}(S^{\mathrm{LCB}}) \lesssim \sum_{j \in S^\star} \sqrt{d \cdot p_j(S^\star|\theta^\star)p_0(S^\star|\theta^\star)\|x_j\|^2_{H_D^{-1}(\theta^\star)}} + d\max_{j \in S^\star}\|x_j\|^2_{H_D^{-1}(\theta^\star)},$$

*with probability at least $1 - \delta$ up to logarithmic factors.*

# 5 Online Linear MNL Bandits

## 5.1 The SupLinearMNL algorithm

In this section, we propose a linear MNL bandit algorithms by leveraging the confidence region result established in Theorem 1. A detailed description of our algorithm is provided in Algorithm 2. To effectively balance the exploration-exploitation trade-off while keeping the independence of collected samples, we apply the SupCB framework of [8], incorporating a sophisticated action set elimination procedure and sample splitting procedure, as in [21, 38, 31, 13, 42]. However, several modifications have been made to our algorithm to ensure the burn-in condition of Theorem 1 and to utilize the first-order geometry of the revenue functions, as described below.

**The SupCB Framework.** Similar to the standard framework in [8], Algorithm 2 divides the collected samples into $S$ bins, denoted as $\Psi_1, \ldots, \Psi_{M+1}$. To maintain independence while accounting for the first-order geometry, we add an additional bin, $\Psi_{S+1}$, as in [31], which contains a rough estimator for approximating the first-order coefficients of $R$. After an initial pure-exploration period of length $\tau$—ensuring that each bin collects sufficient samples, as explained in the next paragraph—the algorithm enters an adaptive elimination phase conducted through a multi-layer procedure that loops over the first $S$ bins.

**Initial Exploration Phase.** The initial exploration phase is designed to ensure the burn-in condition in Theorem 1. More precisely, the initial exploration phase ensures that $\max_{j \in [N]}\|x_j\|_{H_{\tau,\ell}^\dagger(\theta^\star)}$ is well controlled for every $\ell \in [S+1]$ with high probability. Note that since $H_{\tau,\ell}(\theta^\star) \succeq \kappa V_{\tau,\ell}$, with $V_{\tau,\ell} := \sum_{s \leq \tau, s \in \Psi_\ell} \sum_{k \in S_s} x_k x_k^\top$, it suffices to ensure that $\max_{j \in [N]} \kappa^{-1/2}\|x_j\|_{V_{\tau,\ell}^\dagger}$ is well bounded for every $\ell$. To achieve this goal with finite sample complexity, the algorithm repeatedly taking the exploration assortments containing uncertain items until

$$512\kappa^{-1/2}\max_j\|x_j\|_{V_{\tau,\ell}^\dagger} \leq \frac{1}{\sqrt{d\log(NT)}} \wedge \frac{1}{W} \tag{4}$$

holds for all $\ell$. Careful readers may notice that to compute (4), we need prior knowledge of $\kappa$ as an input to Algorithm 2, which is a problem-dependent quantity. When the exact value of $\kappa$ is unknown, we can instead use the parameter radius $W$ (or its upper bound) to provide a conservative estimate. In particular, $\exp(-KW)$ can be used as a worst-case bound for $\kappa$. It should be noted that there may exist problem instances where $\kappa$ is strictly smaller than its worst-case bound. Designing algorithms that achieve the same regret guarantee while fully adapting to an unknown $\kappa$, as in [44, 35], is left for future work.

**Adaptive Elimination Phase.** After entering the adaptive elimination phase, the algorithm stops allocating samples to $\Psi_{S+1}$. The MLE

$$\widehat{\theta}_0 := \mathrm{argmax}_\theta \ell_{D_{\tau,S+1}}(\theta) \tag{5}$$

---

**Algorithm 2** SupLinearMNL

---

1: **Input:** Time horizon $T$, problem-dependent factor $\kappa$.
2: initialize $M = \log_2 T, \lambda = 1, \tau = 1, \Psi_1 = \cdots = \Psi_{M+1} = \emptyset$.
3: **while** (4) is not satisfied for some $\ell \in [M]$ and $j \in [N]$ **do**
4:     Select arbitrary assortment that contains item $j$, add $\tau$ into $\Psi_\ell$
5:     $\tau \leftarrow \tau + 1$
6: **end while**
7: $\Psi_0 \leftarrow \emptyset$, compute $\widehat{\theta}_0$ as in (5)
8: **for** $t = \tau + 1, \ldots, T$ **do**
9:     set $\mathcal{A}_1 = \mathcal{S}_K, S_t = \emptyset, \ell = 1$
10:     **while** $S_t = \emptyset$ **do**
11:         Compute $W_{t,S}^\ell, R_{t,\ell}^{\text{UCB}}(S), \forall S \in \mathcal{A}_\ell$ as in (7), (8).
12:         **if** $W_{t,S}^\ell > 2^{-\ell}$ for some $S \in \mathcal{A}_\ell$ **then**
13:           select such $S \in \mathcal{A}_\ell$.
14:           $\Psi_\ell \leftarrow \Psi_\ell \cup \{t\}$
15:         **else if** $W_{t,S}^\ell \le 1/T$ for all $S \in \mathcal{A}_\ell$ **then**
16:           take the action $S_t = \text{argmax}_{S \in \mathcal{A}_\ell} R_{t,\ell}^{\text{UCB}}(S)$
17:           $\Psi_0 \leftarrow \Psi_0 \cup \{t\}$
18:         **else**
19:           $\widehat{R} \leftarrow \max_{S \in \mathcal{A}_\ell} R_{t,\ell}^{\text{UCB}}(S)$
20:           $\mathcal{A}_{\ell+1} \leftarrow \left\{ S \in \mathcal{A}_\ell, R_{t,\ell}^{\text{UCB}}(S) \ge \widehat{R} - 2^{-\ell+7} \right\}$
21:           $\ell \leftarrow \ell + 1$
22:         **end if**
23:     **end while**
24: **end for**

---

is computed based on the data in $\Psi_{S+1}$ collected during the first $\tau$ rounds and is fixed in all subsequent time steps.

At each $t > \tau$, during the $\ell$-th loop, the algorithm calculates the confidence level term

$$w_{t,i}^\ell := 72\|x_i\|_{(H_{t,\ell}^\lambda)^{-1}(\widehat{\theta}_0)} \sqrt{\log(NT)} \tag{6}$$

for $i \in \cup_{S \in \mathcal{A}_\ell} S$, with $D_{t,\ell}$ the data collected in the $\ell$-th bin up to time $t$ and we simply denote $H_{D_{t,\ell}}^\lambda$ by $H_{t,\ell}^\lambda$. Then, based on the value of $w_{t,i}$ for each $i$, the assortment-wise confidence level for $S$, written as

$$W_{t,S}^\ell := \sum_{i \in S} \sqrt{16 p_i(S|\widehat{\theta}_0) p_0(S|\widehat{\theta}_0)(w_{t,i}^\ell)^2} + \max_{i \in S}(4w_{t,i}^\ell)^2 \tag{7}$$

is calculated for each $S \in \mathcal{A}_\ell$. Then the algorithm takes one of the following steps based on $\{W_{t,S}^\ell\}_,$:

**Step (a). Exploration of Uncertain Items:** If there exists some uncertain assortment (i.e., $W_{t,S}^\ell > 2^{-\ell}$), the algorithm selects this assortment to explore it further.

**Step (b). Output UCB Assortment:** If all assortments are sufficiently certain (i.e., $W_{t,S}^\ell < 1/T$ for all $S \in \mathcal{A}_\ell$), the algorithm outputs the assortment with highest UCB value, computed as

$$R_{t,\ell}^{\text{UCB}}(S) := R(S|\boldsymbol{v}^{\text{UCB}}), \tag{8}$$

with $v_i^{\text{UCB}} := \exp\left(x_i^\top \widehat{\theta}_{D_t,\ell}^\lambda + w_{t,i}^\ell\right), \quad \forall i \in [N]$.

**Step (c). Assortment Elimination:** Otherwise, the algorithm performs assortment elimination over $\mathcal{A}_\ell$. Specifically, it first computes the maximum UCB value $\widehat{R}$. Next, it eliminates all assortments $S$ such that the UCB value gap, $\widehat{R} - R_{t,\ell}^{\text{UCB}}(S)$, exceeds $2^{-\ell}$. Finally, the algorithm loops to the $(\ell + 1)$-th bin with the eliminated item set $\mathcal{A}_{\ell+1}$.

## 5.2 Regret Guarantee of Algorithm 2

Now we show the regret guarantee of Algorithm 2:

**Theorem 3.** *With probability at least $1 - 1/T$, we have Algorithm 2 satisfies*

$$Reg(T) \lesssim \sqrt{dT \log(NT)} \log(T) + \kappa^{-1}(d^2 + dW) \log(dNTW).$$

Theorem 3 establishes a $\widetilde{O}(\sqrt{dT} + \kappa^{-1}d^2K^2)$ regret bound. Compared to the $\widetilde{O}(\kappa^{-1}\sqrt{dT})$ in [42] and $\widetilde{O}(d\sqrt{T} + \kappa^{-1}d^2)$ in [35], our result is the first to achieve $\widetilde{O}(\sqrt{dT \log N})$ regret in the linear MNL setting with a second-order dependence on $\kappa^{-1}$. However, due to the need to enumerate over $\mathcal{A}_\ell$ in the elimination (Line 20) and confidence computation (Line 11) steps—each takes $\Omega(N^K)$ times of computation in worst-case, similar to [18, 42]—the algorithm is computationally inefficient[2]. Thus, Theorem 3 serves primarily as a theoretical benchmark, and we leave designing efficient algorithms with similar guarantees as a future direction.

**Problem-Dependent Regret with Uniform Revenue.** While above results consider the non-uniform revenue setting for general $r \in [0,1]^N$. We can show that in the uniform revenue setting where $r_i \equiv 1$ as in [44, 35], an improved problem-dependent rate is possible: Denote

$$\kappa^\star := \sum_{j \in S^\star} p_j(S^\star|\boldsymbol{v})p_0(S^\star|\boldsymbol{v}) = \frac{\sum_{j \in S^\star} v_j}{(1 + \sum_{j \in S^\star} v_j)^2},$$

we have the following guarantee of Algorithm 2

**Theorem 4.** *In the uniform revenue setting $r_i = 1, \forall i \in [N]$, Algorithm 2 satisfies*

$$Reg(T) \lesssim \sqrt{dT\kappa^\star \log(NT)} \log(T) + \kappa^{-1}(d^2 + dW) \log(dWNT).$$

Noticing that in the good scenario where all $v_i = \Theta(1)$ as in [35], we have the above bound simply turns to a $\widetilde{O}(\sqrt{dT/K})$ result, improving a $\sqrt{K}$ factor than previous best known results. In Appendix, we further show a $\Omega(\kappa^\star\sqrt{dT})$ problem-dependent lower bound for a large range of $\kappa^\star$, implying the optimality of Theorem 4.

**Acknowledgments.** This work is generously supported by NSF CCF-2419564, NSF CCF-2312204, NSF CCF-2312205, ONR-13983263, ONR-13983111, and 2026 New York University Center for Global Economy and Business grant.

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

# A  Technical Overview and Proof of Main Results

## A.1  New Perturbation Results of Revenue Functions

As in previous works, to eliminate the multiplicative dependency on $\kappa^{-1}$, we follow an improved analysis that uses a second-order expansion of the revenue function $R(S \mid e^{\boldsymbol{u}})$ with respect to $\boldsymbol{u} := X^{\top}\theta$. However, when $K \geq 2$, the presence of multiple items in $S$ introduces significant challenges. To handle this, a careful perturbation analysis is required to bound the resulting error. For example, as noted in [35], Equation (16) in [3] incorrectly extends the perturbation result from the logistic bandit($K = 1$) setting, causing their analysis to fail. On the other hand, [44] provides a promising direction for handling such perturbations, though their result only applies to the uniform revenue case where $r_i \equiv 1$. The only promised development for *non-uniform revenue* setting is given by the recent work [35], where an approach based on *centralizing the context features* are provided in the proof of their Theorem 4. However, this argument is somewhat complex and requires a careful auxiliary analysis.

In this section, we present several simpler and general perturbation results for revenue functions for the purpose of analyzing Algorithm 1 and Algorithm 2. As a byproduct, we can show that a straightforward plug-in argument based on our lemma allows us to extend the regret analysis in [44]—which previously applied only to the uniform revenue case—to the general non-uniform revenue setting, achieving the optimal $\widetilde{O}(d\sqrt{T} + \kappa^{-1}d^2)$ regret, we leave the detail to Section E.5.

**Proposition 3.** *For any fixed revenue vector $\boldsymbol{r}$ and utility vectors $\boldsymbol{u}' \in \mathbb{R}^N$, denote $\boldsymbol{u} := \log(\boldsymbol{v})$ and $\boldsymbol{w} := \boldsymbol{u}' - \boldsymbol{u}, \boldsymbol{v}' := e^{\boldsymbol{u}'}$, then*
*i) it holds for any $S_0 \in \mathcal{S}_K$ that*

$$|R(S_0|\boldsymbol{v}') - R(S_0|\boldsymbol{v})| \leq \sqrt{\sum_{j \in S_0} v_j |r_j - R(S_0|\boldsymbol{v})|^2} \cdot \sqrt{\sum_{j \in S_0} p_j(S_0|\boldsymbol{v})p_0(S_0|\boldsymbol{v})w_j^2} + \frac{3}{2}\max_{j \in S_0} w_j^2.$$

*ii) In addition, if $S_0$ satisfies $r_j \geq R(S_0|\boldsymbol{v})$ for all $j \in S_0$, then*

$$|R(S_0|\boldsymbol{v}') - R(S_0|\boldsymbol{v})| \leq \sqrt{\sum_{j \in S_0} p_j(S_0|\boldsymbol{v})p_0(S_0|\boldsymbol{v})w_j^2} + 3\max_{j \in S_0} w_j^2.$$

Based on Proposition 3, we can further present the following result for the analysis of elimination-based Algorithm 2.

**Proposition 4.** *Suppose for some $\widetilde{\boldsymbol{v}} \geq \boldsymbol{v}$ it holds that $R(S_0|\widetilde{\boldsymbol{v}}) \geq R(S^{\star}|\widetilde{\boldsymbol{v}}) - \varepsilon$, then for $\boldsymbol{w} = \widetilde{\boldsymbol{u}} - \boldsymbol{u}$ we have*

$$R(S^{\star}|\boldsymbol{v}) - R(S_0|\boldsymbol{v}) \leq 2\sqrt{\sum_{j \in S_0} p_j(S_0|\boldsymbol{v})p_0(S_0|\boldsymbol{v})w_j^2} + 5\max_{j \in S_0} w_j^2 + 2\varepsilon.$$

## A.2  Proof of Theorem 2

*Proof.* Throughout the proof, we denote $\xi_i := 72\|x_i\|_{H_D^{-1}(\widehat{\theta})}\sqrt{\log(N_{\text{eff}}/\delta)}, \widehat{\theta}_D$ by $\widehat{\theta}$ for simplicity. With above notation, we have

$$R^{\text{LCB}}(S) = R(S|\boldsymbol{v}^{\text{LCB}}) = \frac{\sum_{i \in S} r_i \exp(x_i^{\top}\widehat{\theta} - \xi_i)}{1 + \sum_{i \in S} \exp(x_i^{\top}\widehat{\theta} - \xi_i)}, \quad \forall S \in \mathcal{S}_K.$$

Now we have for $\boldsymbol{v} = \exp(X^{\top}\theta^{\star})$, it holds by Theorem 1 that with probability at least $1 - \delta$ that $\boldsymbol{v}^{\text{LCB}} \leq \boldsymbol{v}$. Under such a condition, we have then

$$R(S^{\star}|\theta^{\star}) - R(S|\theta^{\star}) \leq_{(i)} R(S^{\star}|\boldsymbol{v}) - R(S|\boldsymbol{v}^{\text{LCB}}) \leq_{(ii)} R(S^{\star}|\boldsymbol{v}) - R(S^{\star}|\boldsymbol{v}^{\text{LCB}})$$

$$\leq_{(iii)} \sqrt{\sum_{j \in S^{\star}} p_j(S^{\star}|\boldsymbol{v})p_0(S^{\star}|\boldsymbol{v})w_j^2} + 3\max_{j \in S^{\star}} w_j^2.$$

Where (i) is by the monotone property of $\boldsymbol{v}$ at its the revenue maximizer(see e.g. Lemma A.3 of [5] (ii) is by $R(S^{\star}|\boldsymbol{v}^{\text{LCB}}) \leq R(S|\boldsymbol{v}^{\text{LCB}})$, (iii) is by the statement ii) of Proposition 3 and the fact $r_j \geq R(S^{\star}|\boldsymbol{v}), \forall j \in S^{\star}$. This finishes the proof. □

## A.3 Proof of Theorem 3

We first show the following exploration length upper bound result:

**Lemma 1.** *In Algorithm 2, there exits some absolute constant $c_0$ so that the exploration phase will stop after at most $C_0 M \kappa^{-1} d \left( \sqrt{d \log(NT)} \vee W \right)^2 \log(dNTW)$ iterations, and it holds that*

$$512 \max_{j \in [N]} \|x_j\|_{(H_{\tau,\ell}^\lambda)^{-1}(\theta^\star)} \leq \frac{1}{\sqrt{d \log(NT)}} \wedge \frac{1}{W}$$

*for every $\ell \in [M+1]$.*

In particular, Lemma 1 verifies the burn-in condition in Theorem 1 with $\lambda = 1$. Consequently, it can be applied in subsequent time steps as long as the independence assumption is satisfied, which is guaranteed by the SupCB elimination framework.

Based on our exploration-phase analysis in Lemma 1, we have the exploration phase incurs a regret of order $\widetilde{O}(\kappa^{-1} d^2 K^2 \log(NT))$, and the burn-in condition in Theorem 1 is satisfied for all bins. So it remains to bound the regret incurred from $\tau + 1$ to $T$ under the event

$$|x_j^\top (\widehat{\theta}_{t,\ell}^\lambda - \theta^\star)| \leq (72\sqrt{\log(NT)} + 128W)\|x_j\|_{H_{t,\ell}^{-1}(\theta^\star)}, \quad \forall t \geq \tau, j \in [N], \ell \in [M], \quad (9)$$

and

$$\frac{1}{2} H_{t,\ell}(\theta^\star) \preceq H_{t,\ell}(\widehat{\theta}_0) \preceq 2H_{t,\ell}(\theta^\star), \quad \forall t \geq \tau, j \in [N], \ell \in [M]. \quad (10)$$

which holds with probability at least $1 - O(1/T)$.

We would also note that (9) and (10) together with (4) also implies

$$|x_j^\top (\widehat{\theta}_0 - \theta^\star)| \leq (72\sqrt{\log(NT)} + 128W)\|x_j\|_{H_{\tau,0}^{-1}(\theta^\star)} \leq \frac{72 + 128}{512} \leq 1, \quad \forall j$$

thus for any $S$ and $i \in S$,

$$e^{-2}p_i(S|\widehat{\theta}_0) \leq \frac{e^{\widehat{u}_i - 1}}{1 + \sum_{j \in S} e^{\widehat{u}_j + 1}} \leq p_i(S|\boldsymbol{v}) \leq \frac{e^4 e^{\widehat{u}_i + 1}}{1 + \sum_{j \in S} e^{\widehat{u}_j - 1}} \leq e^2 p_i(S|\widehat{\theta}_0),$$

with $\widehat{u}_i := x_j^\top \widehat{\theta}_0$. As a result,

$$2^{-4} \leq \frac{W_{t,S}^\ell}{\sqrt{\sum_{j \in S} p_j(S|\boldsymbol{v}) p_0(S|\boldsymbol{v}) \|x_j\|_{H_{t,\ell}^{-1}}^2}} \leq 2^4 \quad (11)$$

for any $\ell, S$ and $t \geq \tau$.

**Lemma 2.** *For every $\ell$ and $S \in \mathcal{A}_\ell$ it holds under (9) and (10) that*

$$|R_{t,\ell}^{UCB}(S) - R(S|\theta^\star)| \leq W_{t,S}^\ell + 2^{-\ell+1}.$$

*Proof of Lemma 2.* We divide the proof into two steps:

**Step 1: $S^\star \in \mathcal{A}_\ell$ for each $\ell$.** At each $t \geq \tau$, we first show that $S^\star \in \mathcal{A}_\ell$ for all $\ell$: The claim holds for $\ell = 0$ since $\mathcal{A}_0 = \mathcal{S}_K$. Now suppose $S^\star \in \mathcal{A}_{\ell-1}$ for some $\ell \geq 1$, then suppose the algorithm enters the step (c) at the $\ell$-th loop, it holds that by $\boldsymbol{v}_{t,\ell}^{\text{UCB}} \geq \exp(\boldsymbol{X}^\top \theta^\star)$ under (9) and (10),

$$R_{t,\ell}^{\text{UCB}}(S^\star) \geq R(S^\star|\theta^\star) \geq R(\widehat{S}_\ell|\theta^\star).$$

for $\widehat{S}_\ell := \operatorname{argmax}_{S \in \mathcal{A}_\ell} R_{t,\ell}^{\text{UCB}}(S)$. On the other hand, we cannot directly apply Proposition 3 to obtain a perturbation bound for $\widehat{S}_\ell$, since it maximizes the optimistic revenue over an *unstructured set* $\mathcal{A}_\ell$ rather than the structured set $\mathcal{S}_K$.

For $\widehat{S}_\ell$, it holds that by statement i) of Proposition 3,

$$R_{t,\ell}^{\text{UCB}}(\widehat{S}_\ell) - R(\widehat{S}_\ell|\theta^\star) \leq \sqrt{\sum_{j \in \widehat{S}_\ell} v_j |r_j - R(\widehat{S}_\ell|\boldsymbol{v})|^2} \cdot \sqrt{\sum_{j \in \widehat{S}_\ell} p_j(\widehat{S}_\ell|\boldsymbol{v}) p_0(\widehat{S}_\ell|\boldsymbol{v}) w_j^2} + \frac{3}{2} \max_{j \in \widehat{S}_\ell} w_j^2.$$

Now if we denote

$$\widehat{S}_\ell^+ := \{j \in \widehat{S}_\ell, r_j \geq R(\widehat{S}_\ell | \boldsymbol{v})\}, \widehat{S}_\ell^- := \widehat{S}_\ell \setminus \widehat{S}_\ell^+,$$

then it holds that

$$\sum_{j \in \widehat{S}_\ell} v_j |r_j - R(\widehat{S}_\ell | \boldsymbol{v})|^2 \leq \sum_{j \in \widehat{S}_\ell} v_j |r_j - R(\widehat{S}_\ell | \boldsymbol{v})| = \sum_{j \in \widehat{S}_\ell^+} v_j (r_j - R(\widehat{S}_\ell | \boldsymbol{v})) - \sum_{j \in \widehat{S}_\ell^-} v_j (r_j - R(\widehat{S}_\ell | \boldsymbol{v}))$$

$$= 2 \sum_{j \in \widehat{S}_\ell^+} v_j (r_j - R(\widehat{S}_\ell | \boldsymbol{v})) - R(\widehat{S}_\ell | \boldsymbol{v}) \leq 2 \sum_{j \in \widehat{S}_\ell^+} v_j (r_j - R_{t,\ell}^{\mathrm{UCB}}(\widehat{S}_\ell)) + 2 \sum_{j \in \widehat{S}_\ell^+} v_j (R_{t,\ell}^{\mathrm{UCB}}(\widehat{S}_\ell) - R(\widehat{S}_\ell | \boldsymbol{v}))$$

$$\leq 2 \sum_{j \in \widehat{S}_\ell^+} v_j (r_j - R_{t,\ell}^{\mathrm{UCB}}(\widehat{S}_\ell)) + 2 p_0^{-1}(\widehat{S}_\ell | \boldsymbol{v})(R_{t,\ell}^{\mathrm{UCB}}(\widehat{S}_\ell) - R(\widehat{S}_\ell | \boldsymbol{v}))$$

$$\leq 2 \sum_{j \in \widehat{S}_\ell^+} v_j (r_j - R(S^\star | \boldsymbol{v})) + 2 p_0^{-1}(\widehat{S}_\ell | \boldsymbol{v})(R_{t,\ell}^{\mathrm{UCB}}(\widehat{S}_\ell) - R(\widehat{S}_\ell | \boldsymbol{v}))$$

$$\leq 2 R(S^\star | \boldsymbol{v}) + 2 p_0^{-1}(\widehat{S}_\ell | \boldsymbol{v})(R_{t,\ell}^{\mathrm{UCB}}(\widehat{S}_\ell) - R(\widehat{S}_\ell | \boldsymbol{v})),$$

where in the last second inequality we have used

$$R_{t,\ell-1}^{\mathrm{UCB}}(\widehat{S}_\ell) \geq R_{t,\ell-1}^{\mathrm{UCB}}(S^\star) \geq R(S^\star | \theta^\star) \geq R(\widehat{S}_\ell | \theta^\star),$$

by $S \in \mathcal{A}_\ell$. As a consequence, for $\Delta := R_{t,\ell}^{\mathrm{UCB}}(\widehat{S}_\ell) - R(\widehat{S}_\ell | \theta^\star)$ we get

$$\Delta \leq \sqrt{2 R(S^\star | \boldsymbol{v}) + 2 p_0^{-1}(\widehat{S}_\ell | \boldsymbol{v}) \Delta} \cdot \sqrt{\sum_{j \in \widehat{S}_\ell} p_j(\widehat{S}_\ell | \boldsymbol{v}) p_0(\widehat{S}_\ell | \boldsymbol{v}) w_j^2} + \frac{3}{2} \max_{j \in \widehat{S}_\ell} w_j^2,$$

Using the elementary inequalities

$$z^2 \leq Az + B \implies z \leq \frac{A + \sqrt{A^2 + 4B}}{2} \leq A + \sqrt{B} \implies z^2 \leq 2A^2 + 2B$$

for $A, B, z \geq 0$, we get then

$$\Delta \leq 8 \sum_{j \in \widehat{S}_\ell} p_j(\widehat{S}_\ell | \boldsymbol{v}) w_j^2 + 4 \sqrt{\sum_{j \in \widehat{S}_\ell} p_j(\widehat{S}_\ell | \boldsymbol{v}) p_0(\widehat{S}_\ell | \boldsymbol{v}) w_j^2} + 3 \max_{j \in \widehat{S}_\ell} w_j^2 \leq 2^4 W_{t,\widehat{S}_\ell}^\ell \leq 2^{-\ell+6},$$

where the last second inequality is by (11) and the last inequality is by Algorithm 2 does not enter step (a) in $\ell$-th loop. Now we get

$$R_{t,\ell}^{\mathrm{UCB}}(S^\star) \geq R(\widehat{S}_\ell | \theta^\star) \geq R_{t,\ell}^{\mathrm{UCB}}(\widehat{S}_\ell) - 2^{-\ell+6}$$

thus then $R_{t,\ell}^{\mathrm{UCB}}(S^\star) \in \mathcal{A}_\ell$, as desired.

**Step 2: Bound the regret of assortments in $\mathcal{A}_\ell$**  Noticing that by the selection of $\mathcal{A}_\ell$, we have

$$S \in \mathcal{A}_\ell \implies R_{t,\ell}^{\mathrm{UCB}}(S) \geq R_{t,\ell}^{\mathrm{UCB}}(\widehat{S}_\ell) - 2^{-\ell+7} \geq R_{t,\ell}^{\mathrm{UCB}}(S^\star) - 2^{-\ell+7}$$

$$\implies R(S | \boldsymbol{v}) \geq R(S^\star | \boldsymbol{v}) - W_{t,S}^\ell - 2^{-\ell+7},$$

where in the last line we have used Lemma 4, as desired. $\qquad \square$

Similar to the proofs in [38, 31], we can bound the regret of Algorithm 2 in a layer-wise approach:

**Lemma 3.** *For each round $t > \tau$, let $\ell_t$ denote the value of $\ell$ when $S_t$ is selected. Then, under* (9) *and* (10)*, it holds that*

$$R(S^\star) - R(S_t) \leq \begin{cases} 4 \cdot 2^{-\ell_t+8}, & \text{if } S_t \text{ is selected in step (a),} \\ 4/\sqrt{T}, & \text{if } S_t \text{ is selected in step (b).} \end{cases}$$

*Proof.* We have by $S_t \in \mathcal{A}_{\ell_t}$,

$$R(S_t|\theta^\star) \geq R^{\text{UCB}}_{t,\ell_t-1}(S_t) - W^{\ell_t-1}_{t,S_t} - 2^{-\ell_t+8} \geq \widehat{R} - 2^{-\ell_t+9}$$
$$\geq R^{\text{UCB}}_{t,\ell_t-1}(S^\star) - 2^{-\ell_t+9} \geq R(S^\star|\theta^\star) - 2^{-\ell_t+9},$$

this shows the first inequality.

To show the second inequality, noticing that by Lemma 2, the condition of step (b) implies

$$|R(S) - R^{\text{UCB}}_{t,\ell_t}(S)| \leq 8/T, \quad \forall S \in \mathcal{A}_{\ell_t}.$$

Then by

$$R(S_t) \geq R^{\text{UCB}}_{t,\ell_t}(S_t) - 8/T \geq R^{\text{UCB}}_{t,\ell_t}(S^\star) - 8/T \geq R(S^\star) - 1 = 8/T,$$

the desired result holds. $\qquad\square$

Now with Lemma 3, we can bounding the regret of Algorithm 2 for each bin $\Psi_\ell$ separately. More precisely, we have

**Lemma 4.** *For each $\ell \in [S]$, we have condition on $\mathcal{E}_1, \mathcal{E}_2$,*

$$\sum_{t \in \Psi_\ell, t > \tau} R(S^\star) - R(S_t) \lesssim \sqrt{dT \log(NT)} + \frac{d}{\kappa} \log(NT). \tag{12}$$

*Proof.* We divide the time indices in $\Psi_\ell$ into

$$\mathcal{T}_{a,\ell} := \{t \in \Psi_\ell, t > \tau, S_t \text{ is selected at step } (a)\},$$
$$\mathcal{T}_{b,\ell} := \{t \in \Psi_\ell, t > \tau, S_t \text{ is selected at step } (b)\}.$$

By Lemma 3 and $2^{-\ell} \leq W^\ell_{t,S_t}$ for all $j \in S_t$ and $t \in \mathcal{T}_a$, we have

$$\sum_{t \in \mathcal{T}_{a,\ell}} R(S^\star) - R(S_t) \leq \sum_{t \in \mathcal{T}_{a,\ell}} 4 \cdot 2^{-\ell+9}$$
$$\lesssim \sum_{t \in \mathcal{T}_{a,\ell}} \Big[ \sum_{j \in S_t} \sqrt{p_j(S_t)p_0(S_t)\|x_j\|^2_{H^{-1}_{t,\ell}(\theta^\star)} \log NT} + \max_{j \in S_t}(72\sqrt{\log NT}\|x_j\|_{H^{-1}_{t,\ell}(\theta^\star)})^2 \Big] \tag{13}$$

To bound the above summation over $\mathcal{T}_{a,\ell}$, we apply the following linear MNL version elliptical potential lemma:

**Lemma 5** (Lemma E.2 in [35]). *For $\lambda \geq 1$, it holds that*
1. $\sum_{t \in \mathcal{T}_{a,\ell}} \sum_{j \in S_t} p_j(S_t)p_0(S_t)\|x_j\|^2_{H^{-1}_{t,\ell}} \leq 2d \log(1 + \frac{|\mathcal{T}_{a,\ell}|}{2d\lambda})$.
2. $\sum_{t \in \mathcal{T}_{a,\ell}} \max_{i \in S_t} \|x_t\|^2_{H^{-1}_{t,\ell}} \leq 2d\kappa^{-1} \log(1 + \frac{|\mathcal{T}_{a,\ell}|}{d\lambda})$.

Applying this Lemma and Cauchy-Schwartz inequality in (13) then leads to

$$\sum_{t \in \mathcal{T}_{a,\ell}} R(S^\star) - R(S_t) \lesssim \sqrt{d|\mathcal{T}_{a,\ell}| \log(NT) \log(1 + T/d)} + \frac{2d}{\kappa} \log(1 + T/d) \log(NT),$$

this provides the regret bound for the summation over $\mathcal{T}_{a,\ell}$.

On the other hand, by Lemma 2 we have taking summation over $t \in \mathcal{T}_{b,\ell}$ naturally leads to

$$\sum_{t \in \mathcal{T}_{b,\ell}} R(S^\star) - R(S_t) \leq 8 \log T.$$

combining the upper bounds for $\mathcal{T}_{a,\ell}, \mathcal{T}_{b,\ell}$ and taking summation over $\ell$ then finishes the proof. $\quad\square$

# B   Proof of Theorem 1

We begin with the following Lemma:

**Lemma 6.** *Suppose the same independence structure as in Theorem 1 and denote*

$$\zeta := 3\sqrt{2} \max_{m \leq t, j \in S_m} |x_{m,j}^\top (\widehat{\theta}_D^\lambda - \theta^\star)|, \quad \xi := \max_{m \leq t, j \in S_m} \|x_{m,j}\|_{(H_t^\lambda)^{-1}}, \quad \varphi(\zeta) := (\frac{e^\zeta - 1}{\zeta} - 1)(1 + \zeta)$$

*we have*

$$\frac{|x^\top (\widehat{\theta}_D^\lambda - \theta^\star)|}{8\|x\|_{(H_t^\lambda)^{-1}}} \leq \left(\sqrt{\log(1/\delta)} + \xi \log(1/\delta)\right) + \varphi(\zeta) \left(\sqrt{d \log(1/\delta)} + dK \log(1/\delta) \cdot \xi + \sqrt{\lambda}W\right) + \sqrt{\lambda(1+\zeta)}W.$$

*Proof of Lemma 6.* Since in this formulation the feature vector $x_{m,j}$ may change for each $m$, we introduce the notation $p_{mj}(\theta)$ to represent the probability that item $j$ is chosen given $S_m$ and $\boldsymbol{x}_m$. We also denote $H_D^\lambda(\theta^\star)$ by $H_t^\lambda$, $H_D(\theta^\star)$ by $H_t$ for simplicity.

Noticing that by $\widehat{\theta}_D^\lambda$ maximizes $\ell_D^\lambda$, we have

$$\nabla \ell_D^\lambda(\widehat{\theta}_D^\lambda) = 0 \implies \sum_{m=1}^t \sum_{j \in S_m} x_{m,j} \left(p_{mj}(\widehat{\theta}_D^\lambda) \pm p_{mj}(\theta^\star) - y_{mj}\right) + \lambda \widehat{\theta}_D^\lambda = 0 \implies$$

$$\sum_{m=1}^t \sum_{j \in S_m} x_{m,j} \left(p_{mj}(\widehat{\theta}_D^\lambda) - p_{mj}(\theta^\star)\right) + \lambda(\widehat{\theta}_D^\lambda - \theta^\star) = \sum_{m=1}^t \sum_{j \in S_m} x_{m,j} \underbrace{(y_{mj} - p_{mj}(\theta^\star))}_{:=\eta_{m,j}} - \lambda\theta^\star$$

Now if we denote

$$L_m(\theta) := \sum_{j \in S_m} x_{m,j} p_{mj}(\theta),$$

then for any $\theta$ it holds that

$$\sum_{j \in S_m} x_{m,j} \left(p(j; S_m, \boldsymbol{x}_m, \theta) - p_{mj}(\theta^\star)\right)$$

$$= L_m(\theta) - L_m(\theta^\star) = \int_0^1 DL_m(\theta^\star + s(\theta - \theta^\star))(\theta - \theta^\star)ds$$

$$= DL_m(\theta^\star)(\widehat{\theta}_D^\lambda - \theta^\star) + \left(\int_0^1 DL_m(\theta^\star + s(\theta - \theta^\star)) - DL_m(\theta^\star)ds\right)(\theta - \theta^\star),$$

with

$$DL_m(\theta) := \sum_{j \in S_m} p_{mj}(\theta) x_{mj} x_{mj}^\top - \sum_{i \in S_m, j \in S_m} p_{mj}(\theta) p_{mi}(\theta) x_{mi} x_{mj}^\top.$$

Now notice that $H_t^\lambda = \sum_{m=1}^t DL_m(\theta^\star) + \lambda I$ and for

$$E_t := \sum_{m=1}^t \int_0^1 DL_m(\theta^\star + s(\widehat{\theta}_D^\lambda - \theta^\star)) - DL_m(\theta^\star)ds,$$

we have

$$x^\top (\widehat{\theta}_D^\lambda - \theta^\star)$$

$$= x^\top (H_t + E_t + \lambda I)^{-1} \left(\sum_{m=1}^t \sum_{j \in S_m} x_{m,j} \eta_{m,j} - \lambda\theta^\star\right)$$

$$= x^\top \left((H_t^\lambda)^{-1} - (H_t^\lambda)^{-1} E_t (H_t^\lambda + E_t)^{-1}\right) \left(\sum_{m=1}^t \sum_{j \in S_m} x_{m,j} \eta_{m,j} - \lambda\theta^\star\right)$$

$$= \underbrace{x^\top (H_t^\lambda)^{-1} \sum_{m=1}^t \sum_{j \in S_m} x_{m,j} \eta_{m,j}}_{:=J_1} - \underbrace{x^\top (H_t^\lambda)^{-1} E_t (H_t^\lambda + E_t)^{-1} \sum_{m=1}^t \sum_{j \in S_m} x_{m,j} \eta_{m,j}}_{:=J_2} - \underbrace{\lambda x^\top (H_t^\lambda + E_t)^{-1}\theta^\star}_{:=J_3}$$

$$(14)$$

**The first term $J_1$**

The first term can be bounded by variance-aware concentration results using conditional independence as in previous works obtaining sharp bounds for logistic setting [31] , the main difference between the logistic setting is the dependency across $\eta_{m,j}$ for $j \in S_m$ :

For every $m$, we have

$$Z_m := \sum_{j \in S_m} x^\top (H_t^\lambda)^{-1} x_{m,j} \eta_{m,j}$$

is centered and bounded as

$$|Z_m| \leq \max_j |x^\top (H_t^\lambda)^{-1} x_{m,j}| \cdot \sum_{j \in S_m} |\eta_{m,j}| \leq 2 \max_j |x^\top (H_t^\lambda)^{-1} x_{m,j}|.$$

With variance

$$\mathbb{E}[Z_m^2] = \sum_{i,j \in S_m} (x^\top (H_t^\lambda)^{-1} x_{m,j})(x^\top (H_t^\lambda)^{-1} x_{m,i}) \mathbb{E}[\eta_{m,j} \eta_{m,i}].$$

By

$$\mathbb{E}[\eta_{m,i}^2] = p_{mi}(\theta^\star)(1 - p_{mi}(\theta^\star)), \mathbb{E}[\eta_{m,i} \eta_{m,j}] = -p_{mi}(\theta^\star) p_{mj}(\theta^\star),$$

we have

$$\mathbb{E}[Z_m^2] = x^\top (H_t^\lambda)^{-1} \underbrace{\left( \sum_{i \in S_m} x_{m,i} x_{m,i}^\top p_{mi}(\theta^\star)(1 - p_{mi}(\theta^\star)) - \sum_{i \neq j} x_{m,i} x_{m,j}^\top p_{mi}(\theta^\star) p_{mj}(\theta^\star) \right)}_{:=U_m} (H_t^\lambda)^{-1} x,$$

now by $\sum_{m=1}^t U_m = H_t$, we have $\sum_m \mathbb{E}[Z_m^2] = x^\top (H_t^\lambda)^{-1} H_t (H_t^\lambda)^{-1} x \leq x^\top (H_t^\lambda)^{-1} x$, now we can apply the following Bernstein inequality:

**Lemma 7** (Bernstein's Inequality). *Let $X_1, \ldots, X_n$ be independent zero-mean random variables. Suppose that $|X_i| \leq M$ almost surely, for all $i$. Then, for all positive $t$,*

$$\mathbb{P}\left( \sum_{i=1}^n X_i \geq u \right) \leq \exp\left( -\frac{\frac{1}{2} u^2}{\sum_{i=1}^n \mathbb{E}\left[ X_i^2 \right] + \frac{1}{3} M u} \right).$$

to obtain that

$$\mathbb{P}\left( |J_1| \geq u \right) \leq \exp\left( -\frac{u^2}{2\|x\|_{(H_t^\lambda)^{-1}}^2 + 2 \max_{m \leq t} \max_{j \in m} |x^\top (H_t^\lambda)^{-1} x_{m,j}| u} \right),$$

or equivalently, with probability at least $1 - \delta$,

$$|J_1| \leq 2\|x\|_{(H_t^\lambda)^{-1}} \left( \sqrt{\log(1/\delta)} + \max_{m \leq t, j \in S_m} \|x_{m,j}\|_{(H_t^\lambda)^{-1}} \log(1/\delta) \right). \tag{15}$$

**The second term $J_2$**

For the second term, we have denote $z_t := \sum_{m=1}^t \sum_{j \in S_m} x_{m,j} \eta_{m,j}$, then

$$|J_2| = |x^\top (H_t^\lambda)^{-1} E_t (H_t^\lambda + E_t)^{-1} z_t|$$
$$\leq \|x\|_{(H_t^\lambda)^{-1}} \|(H_t^\lambda)^{-1/2} E_t (H_t^\lambda)^{-1/2}\| \|(H_t^\lambda + E)^{-1} z_t\|_{H_t^\lambda}).$$

**Bounding $\|(H_t^\lambda)^{-1} E_t (H_t^\lambda)^{-1}\|$.** Given any vector $w$, if we denote

$$a_{mj} := x_{mj}^\top w, b_{mj} = x_{mj}^\top (\widehat{\theta}_D^\lambda - \theta^\star), p_{mj}(s) := p_{mj}(\theta^\star + s(\widehat{\theta}_D^\lambda - \theta^\star)),$$

and

$$u_{mj}(s) := x_{mj}^\top (\theta^\star + s(\widehat{\theta}_D^\lambda - \theta^\star)), v_{mj}(s) := e^{u_{mj}(s)}$$

for $j \in S_m$ and set $a_{m0} = b_{m0} = u_{m0} = 0, v_{m0} = 1$ for convenience. Then we have as shown in [35],

$$\|w\|_{DL_m(\theta^\star + s(\widehat{\theta}_D^\lambda - \theta^\star))}^2 = \frac{\sum_{j \in S_m} \sum_{i \in S_m : 0 \le i < j} (a_{mi} - a_{mj})^2 v_{mi}(s) v_{mj}(s)}{(1 + \sum_{j \in S_m} v_{mj}(s))^2}.$$

It can be verified by calculation that

$$\left| \frac{d}{ds} \|w\|_{DL_m(\theta^\star + s(\widehat{\theta}_D^\lambda - \theta^\star))}^2 \right|$$

$$= \left| \frac{\sum_{j \in S_m} \sum_{i \in (S_m)_+ : 0 \le i < j} (a_{mi} - a_{mj})^2 v_{mi}(s) v_{mj}(s) \left[ \sum_{k \in (S_m)_+} (b_{mi} + b_{mj} - 2b_{mk}) v_{mk}(s) \right]}{(1 + \sum_{j \in S_m} v_{mj}(s))^3} \right|$$

$$\le 3\sqrt{2} \max_{j \in S_m} |b_{mj}| \cdot \frac{\sum_{j \in S_m} \sum_{i \in (S_m)_+ : 0 \le i < j} (a_{mi} - a_{mj})^2 v_{mi}(s) v_{mj}(s)}{(1 + \sum_{j \in S_m} v_{mj}(s))^2}$$

$$\le 3\sqrt{2} \max_{j \in S_m} |b_{mj}| \|w\|_{DL_m(\theta^\star + s(\widehat{\theta}_D^\lambda - \theta^\star))}^2.$$

As a result, $\psi(s) := \log \|w\|_{DL_m(\theta^\star + s(\widehat{\theta}_D^\lambda - \theta^\star))}^2$ satisfies $|\psi'(s)| \le 3\sqrt{2} \max_{j \in S_m} |b_{mj}|$, which then leads to

$$e^{-3\sqrt{2} \max_{j \in S_m} |b_{mj}| s} \|w\|_{DL_m(\theta^\star)} \le \|w\|_{DL_m(\theta^\star + s(\widehat{\theta}_D^\lambda - \theta^\star))} \le e^{3\sqrt{2} \max_{j \in S_m} |b_{mj}| s} \|w\|_{DL_m(\theta^\star)}. \tag{16}$$

Now we have for any $w \in \mathbb{R}^d$, by taking summation over $m$,

$$w^\top E_t w = w^\top \int_0^1 \sum_{m \le t} \left( DL_m(\theta^\star + s(\widehat{\theta}_D^\lambda - \theta^\star)) - DL_m(\theta^\star) \right) ds \, w$$

$$\le \int_0^1 (e^{\zeta s} - 1) ds \|w\|_{H_t}^2 \le \left( \frac{e^\zeta - 1}{\zeta} - 1 \right) \|w\|_{H_t}^2,$$

this leads to

$$E_t \preceq \left( \frac{e^\zeta - 1}{\zeta} - 1 \right) H_t \preceq \left( \frac{e^\zeta - 1}{\zeta} - 1 \right) H_t^\lambda. \tag{17}$$

On the other hand, we have by

$$H_t + E_t = \int_0^1 \sum_{m \le t} DL_m(\theta^\star + s(\widehat{\theta}_D^\lambda - \theta^\star)) ds$$

$$\succeq \int_0^1 e^{-\zeta s} ds \cdot H_t \succeq \frac{1 - e^{-\zeta}}{\zeta} H_t \succeq \frac{1}{1 + \zeta} H_t, \tag{18}$$

where the last inequality is by the elementary inequality $\frac{1 - e^{-x}}{x} \ge \frac{1}{1+x}, x > 0$. With (17) and (18), we arrive at

$$\|(H_t^\lambda)^{-1/2} E_t (H_t^\lambda)^{-1/2}\| = \max_{\|w\|_2 = 1} |w^\top (H_t^\lambda)^{-1/2} E_t (H_t^\lambda)^{-1/2} w|$$

$$\le \left\{ \frac{e^\zeta - 1}{\zeta} - 1, \frac{1}{1 + \zeta} - 1 \right\} \le 2 \left( \frac{e^\zeta - 1}{\zeta} - 1 \right),$$

and thus

$$J_2 \le 2\|x\|_{(H_t^\lambda)^{-1}} \left( \frac{e^\zeta - 1}{\zeta} - 1 \right) \|(H_t^\lambda + E)^{-1} z_t\|_{H_t^\lambda}.$$

**Bounding** $\|(H_t^\lambda + E)^{-1}z_t\|_{H_t^\lambda}$**:** By (18), we have

$$H_t \preceq (1+\zeta)(H_t + E_t) \implies (H_t^\lambda + E_t)^{-1}H_t^\lambda(H_t^\lambda + E_t)^{-1} \preceq (1+\zeta)(H_t^\lambda + E)^{-1} \preceq (1+\zeta)^2 H_t^\lambda.$$

thus it holds that

$$\|(H_t^\lambda + E)^{-1}z_t\|_{H_t^\lambda}^2 = z_t^\top(H_t^\lambda + E_t)^{-1}H_t^\lambda(H_t^\lambda + E_t)^{-1}z_t$$
$$\leq (1+\zeta)^2\|z_t\|_{(H_t^\lambda)^{-1}}^2$$

To control $\|z_t\|_{(H_t^\lambda)^{-1}}$, we have for any unit vector $w$, it holds that

$$w^\top(H_t^\lambda)^{-1/2}z_t = \underbrace{((H_t^\lambda)^{1/2}w)^\top}_{:=\widetilde{x}^\top}(H_t^\lambda)^{-1}z_t,$$

which has the same form as $J_1$ when we replace $\widetilde{x}$ by $x$, thus by the same argument as in bounding $J_1$, we have (15) still holds: with probability at least $1 - \delta'$,

$$|w^\top(H_t^\lambda)^{-1/2}z_t| \leq 2\underbrace{\|\widetilde{x}\|_{(H_t^\lambda)^{-1}}}_{=\|w\|_2=1}\left(\sqrt{\log(1/\delta')} + \max_{m\leq t,j\in S_m}\|x_{m,j}\|_{(H_t^\lambda)^{-1}}\log(1/\delta')\right),$$

now taking $w$ over the $1/2$-net of the unit ball and taking union bound with selecting $\delta' \asymp 4^d\delta$ leads to with probability at least $1 - \delta$,

$$\|z_t\|_{(H_t^\lambda)^{-1}} \leq 8\sqrt{d\log(1/\delta)} + 8d\log(1/\delta) \cdot \max_{m\leq t,j\in S_m}\|x_{m,j}\|_{(H_t^\lambda)^{-1}}.$$

Further introduce the notation $\xi = \max_{m\leq t,j\in S_m}\|x_{m,j}\|_{(H_t^\lambda)^{-1}}$ in Lemma 6, we have then

$$\|z_t\|_{(H_t^\lambda)^{-1}} \leq (1+\zeta)(8\sqrt{d\log(1/\delta)} + 8d\xi\log(1/\delta)) \tag{19}$$

Now we arrive at

$$J_2 \leq 8\|x\|_{(H_t^\lambda)^{-1}}\underbrace{\left(\frac{e^\zeta - 1}{\zeta} - 1\right)(1+\zeta)\left(\sqrt{d\log(1/\delta)} + d\log(1/\delta)\cdot\xi\right)}_{=\varphi(\zeta)}.$$

**Remark 1.** *In the above proof of* (17)*, we borrow the calculation in [35], which was used to verify the self-concordant-like property of the linear MNL-likelihood function. The key distinction in our approach is that we retain the $b_{mj}$ term throughout the calculation, whereas [35] directly apply the bound $|b_{mj}| \lesssim \|\widehat{\theta}_D^\lambda - \theta^\star\|_2$. This difference allows us to derive a bound on $E_t$ that depends only on $\max_{m,j}|x_{mj}^\top(\widehat{\theta}_D^\lambda - \theta^\star)|$, rather than on $\|\widehat{\theta}_D^\lambda - \theta^\star\|_2$. It is worth noting that a very recent work [36] also uses a similar argument to establish a refined self-concordant-like property for the MNL likelihood function (specifically, the $\ell_\infty$ self-concordant-like property in their Proposition B.3) to obtain sharper confidence bounds, and the bound in* (17) *can also be derived from their Proposition B.3.*

*While the above proof primarily focuses on the relation between $H_D(\widehat{\theta}_D^\lambda)$ and $H_D(\theta^\star)$, the inequality in* (16) *can also imply a dominance relation between $H_D(\theta)$ and $H_D(\theta^\star)$ for general $\theta$. We summarize this general result in the following proposition for future applications.*

**Proposition 5.** *Given any $\theta \in \mathbb{R}^d, \lambda \geq 0$, denoting $\zeta_\theta := 3\sqrt{2}\max_{m\leq t,j\in S_m}|x_{m,j}^\top(\theta - \theta^\star)|$, then it holds that*

$$\frac{1}{1 + 2\varphi(\zeta_\theta)}H_D(\theta^\star) \preceq H_D(\theta) \preceq (1 + 2\varphi(\zeta_\theta))H_D(\theta^\star).$$

*Proof of Proposition 5.* By taking $s = 1$ in (16), we can get

$$e^{-\zeta_\theta}\|w\|_{H_D(\theta^\star)} \leq \|w\|_{H_D(\theta)} \leq e^{\zeta_\theta}\|w\|_{H_D(\theta^\star)}$$

for any unit vector $w$, thus it holds that

$$e^{-\zeta_\theta}H_D(\theta^\star) \preceq H_D(\theta) \preceq e^{\zeta_\theta}H_D(\theta^\star).$$

Now by

$$1 + 2\varphi(\zeta_\theta) = 1 + 2(1+\zeta_\theta)(\frac{e^{\zeta_\theta} - 1}{\zeta_\theta} - 1) = e^{\zeta_\theta} + (e^{\zeta_\theta} - 1 - \zeta_\theta) + 2(\frac{e^{\zeta_\theta} - 1}{\zeta_\theta} - 1 - \frac{\zeta_\theta}{2}) \geq e^{\zeta_\theta},$$

the claim holds. $\qquad\square$

**The last term $J_3$**

For the last term in (14), we have by

$$H_t + E_t + \lambda I \succeq \frac{1}{1+\zeta} H_t + \lambda I \succeq \frac{1}{1+\zeta} H_t^\lambda$$

$$J_3 \leq |\lambda x^\top (H_t^\lambda + E_t)^{-1} \theta^\star| \leq \sqrt{\lambda} \|\theta^\star\| \|x\|_{(H_t^\lambda + E_t)^{-1}} \leq \sqrt{\lambda(1+\zeta)} W \|x\|_{(H_t^\lambda)^{-1}}.$$

**Combining all bounds**

Now combining all above bounds, we get

$$\frac{|x^\top (\widehat{\theta}_D^\lambda - \theta^\star)|}{8\|x\|_{(H_t^\lambda)^{-1}}} \leq \left(\sqrt{\log(1/\delta)} + \xi \log(1/\delta)\right) + \varphi(\zeta)\left(\sqrt{d\log(1/\delta)} + d\log(1/\delta)\cdot\xi + \sqrt{\lambda}W\right) + \sqrt{\lambda(1+\zeta)}W,$$

as desired. $\qquad\qquad\qquad\qquad\qquad\qquad\qquad\qquad\qquad\qquad\qquad\qquad\qquad\qquad\qquad\square$

Lemma 6 provides an upper bound on the ratio between $x^\top (\widehat{\theta}_D^\lambda - \theta^\star)$ and $\|x\|_{(H^\lambda)^{-1}}$. However, the upper bound involves both $\xi$ and $\zeta$ simultaneously. Our next lemma shows that when $\xi$ is well-controlled, the $\zeta$ factor can be bounded by $\xi$.

**Lemma 8.** *Under the condition*

$$\xi \leq \min\left\{\frac{1}{64\sqrt{d\log(N_{\mathit{eff}}/\delta)}}, \frac{1}{64\sqrt{\lambda}W}\right\},$$

*we have*

$$\varphi(\zeta) \leq \zeta \leq 16\left(\sqrt{\log(N_{\mathit{eff}}/\delta)} + \sqrt{2\lambda}W\right)\xi + 16\xi^2 \log(N_{\mathit{eff}}/\delta). \tag{20}$$

*Proof.* Applying Lemma 6 to each $x_{m,j}$, we have then with probability at least $1 - \delta$,

$$\frac{\zeta}{8\xi} \leq \left(\sqrt{\log(N_{\text{eff}}/\delta)} + \xi\log(N_{\text{eff}}/\delta)\right) + \varphi(\zeta)\left(\sqrt{d\log(N_{\text{eff}}/\delta)} + d\log(N_{\text{eff}}/\delta)\cdot\xi + \sqrt{\lambda}W\right) + \sqrt{\lambda(1+\zeta)}W$$

Now by the elementary inequality

$$e^x \leq 1 + x + \frac{5}{8}x^2, \forall x \in [0, \frac{3}{5}],$$

it holds that

$$\zeta \leq \frac{3}{5} \implies (1+\zeta)\left(\frac{e^\zeta - 1}{\zeta} - 1\right) \leq \frac{8}{5}\cdot\frac{5\zeta}{8} \leq \zeta,$$

we get when $\zeta \leq \frac{3}{5}$, the above inequality can be reduced to

$$\left(1 - 8\xi(\sqrt{d\log(N_{\text{eff}}/\delta)} + d\xi\log(N_{\text{eff}}/\delta) + \sqrt{\lambda}W)\right)\zeta \leq 8(\sqrt{\log(N_{\text{eff}}/\delta)} + \sqrt{2\lambda}W)\xi + 8\xi^2\log(N_{\text{eff}}/\delta).$$

As a result, if it holds simultaneously that

$$\zeta \leq \frac{3}{5}, \xi \leq \min\left\{\frac{1}{32\sqrt{d\log(N_{\text{eff}}/\delta)}}, \frac{1}{32\sqrt{\lambda}W}\right\},$$

then

$$\zeta \leq 16(\sqrt{\log(N_{\text{eff}}/\delta)} + \sqrt{2\lambda}W)\xi + 16\xi^2\log(N_{\text{eff}}/\delta).$$

On the other hand, by (19), it always holds that with probability at least $1 - \delta$,

$$\begin{aligned}
\zeta \leq \xi\|\widehat{\theta}_D^\lambda - \theta^\star\|_{H_t^\lambda} &= \xi\|(H_t^\lambda + E)^{-1}(z_t - \lambda\theta^\star)\|_{H_t^\lambda} \\
&\leq \xi(1+\zeta)\|(H_t^\lambda + E)^{-1}(z_t - \lambda\theta^\star)\|_{(H_t^\lambda)^{-1}} \\
&\leq \xi(1+\zeta)\left(8\sqrt{d\log(1/\delta)} + 8d\xi\log(1/\delta) + \sqrt{\lambda}W\right)
\end{aligned}$$

As a consequence,

$$\xi \le \min\left\{\frac{1}{64\sqrt{d\log(1/\delta)}}, \frac{1}{64\sqrt{\lambda}W}\right\} \implies \zeta \le 16\xi(\sqrt{d\log(1/\delta)} + d\xi\log(1/\delta) + \sqrt{\lambda}W/16)$$

$$\implies \zeta \le \frac{1}{4} + \frac{1}{64} + \frac{1}{4} \le 3/5.$$

That verifies $\xi \le \min\left\{\frac{1}{64\sqrt{d\log(N_{\text{eff}}/\delta)}}, \frac{1}{64\sqrt{\lambda}W}\right\}$ is sufficient for (20) holds, as desired. $\qquad\square$

Now combining the results in Lemma 6 and Lemma 8, we have $\xi \le \min\left\{\frac{1}{64\sqrt{d\log(N_{\text{eff}}/\delta)}}, \frac{1}{64\sqrt{\lambda}W}\right\}$ implies that

$$\frac{|x^\top(\widehat{\theta}_D^\lambda - \theta^\star)|}{8\|x\|_{(H_t^\lambda)^{-1}}}$$

$$\le \varphi(\zeta)\left(\sqrt{d\log(1/\delta)} + d\log(1/\delta)\cdot\xi + \sqrt{\lambda}W\right) + \sqrt{\lambda(1+\zeta)}W + \sqrt{\log(1/\delta)} + \xi\log(1/\delta)$$

$$\le 16\xi\left((1 + \frac{1}{64\sqrt{d}})\sqrt{\log(N_{\text{eff}}/\delta)} + \sqrt{2\lambda}W\right)\cdot\left(\sqrt{d\log(1/\delta)} + d\log(1/\delta)\cdot\xi + \sqrt{\lambda}W\right)$$

$$+ \sqrt{2\lambda}W + \sqrt{\log(1/\delta)} + \xi\log(1/\delta)$$

$$\le \left((1 + \frac{1}{64\sqrt{d}})\sqrt{\log(N_{\text{eff}}/\delta)} + \sqrt{2\lambda}W\right)\cdot\left(16 + \frac{1}{256} + \frac{1}{4}\right) + \sqrt{\lambda(1+\zeta)}W + 2\sqrt{\log(1/\delta)}$$

$$\le 18\sqrt{2\log(N_{\text{eff}}/\delta)} + 32\sqrt{2\lambda}W.$$

This finishes the proof of the confidence bound result under the burn-in condition.

Moreover we have by $\varphi(\zeta) \le \zeta \le 3/5$, Proposition 5 implies

$$\frac{1}{3}H_t^\lambda \preceq \frac{1}{1+2\zeta}H_t^\lambda \preceq H_t^\lambda + E_t \preceq (1 + 2\varphi(\zeta))H^\lambda \preceq 3H_t^\lambda.$$

This finishes the proof of Theorem 1.

## C Perturbation Results for Revenue Functions

Before proving results in Section 4 and Section 5, we first introduce several perturbation results on the revenue function $R$ with respect to the utility function $\boldsymbol{u}$.

To emphasis its dependency on the utility variable $\boldsymbol{u}$, we use the following notation throughout this section for given $S$ and $\boldsymbol{r}$:

$$Q(\boldsymbol{u}; \boldsymbol{r}, S) := R(S|e^{\boldsymbol{u}}) = \frac{\sum_{j\in S} r_j e^{u_j}}{1 + \sum_{j\in S} e^{u_j}}. \tag{21}$$

Since $\boldsymbol{r}$ is fixed throughout the whole paper, we denote $Q(\boldsymbol{u}; \boldsymbol{r}, S)$ by $Q(\boldsymbol{u}; S)$ for simplicity. We first recall the following first-order and second-order derivative results, originally proved in [44] and refined in [35], we provide the detailed proof only for completeness.

**Proposition 6** (Lemma E.3 of [35]). *Given $\boldsymbol{r} \in [0,1]^N$ and $S \subset [N]$ fixed, if we denote*

$$\boldsymbol{w}_S := \begin{cases} w_j, & j \in S \\ 0, & j \notin S \end{cases} \tag{22}$$

*for $\boldsymbol{w} \in \mathbb{R}^N$. Then for any $\boldsymbol{u} \in \mathbb{R}^N$, we have*

$$\langle \nabla Q(\boldsymbol{u}; S), \boldsymbol{w}\rangle = \sum_{i\in S} p_i(S|e^{\boldsymbol{u}})(r_i - Q(\boldsymbol{u}; S))w_i, \tag{23}$$

$$\left|\boldsymbol{w}^\top \nabla^2 Q(\boldsymbol{u}; S)\boldsymbol{w}\right| \le 3\max_{i\in S} w_i^2. \tag{24}$$

*Proof of Proposition 6.* During the proof $S$ is fixed, thus we simply omit the notation and simply denote $Q(\boldsymbol{u}; S)$ as $Q(\boldsymbol{u})$.

To prove (23), noticing that for every $i \notin S, \partial_{u_i} Q(\boldsymbol{u}) = 0$ and for every $i \in S$, we have

$$
\begin{aligned}
\partial_{u_i} Q(\boldsymbol{u}) &= \frac{r_i e^{u_i}}{1 + \sum_{j \in S} e^{u_j}} - \frac{e^{u_i} \sum_{j \in S} r_j e^{u_j}}{(1 + \sum_{j \in S} e^{u_j})^2} \\
&= p_i(S|e^{\boldsymbol{u}})\big(r_i - Q(\boldsymbol{u})\big)
\end{aligned}
$$

Thus

$$
\langle \nabla Q(\boldsymbol{u}), \boldsymbol{w} \rangle = \sum_{i \in S} \partial_{u_i} Q(\boldsymbol{u}) w_i = \sum_{i \in S} p_i(S|e^{\boldsymbol{u}})\big(r_i - Q(\boldsymbol{u})\big) w_i,
$$

as desired.

To prove (24), noticing that $\partial^2_{u_i, u_j} Q(\boldsymbol{u}) = 0$ if $i \notin S$ or $j \notin S$. When $i, j \in S$, we have

$$
\begin{aligned}
\partial_{u_j} \partial_{u_i} Q(\boldsymbol{u}) &= \partial_{u_j}\big[p_i(S|e^{\boldsymbol{u}})(r_i - Q(\boldsymbol{u}))\big] \\
&= \partial_{u_j} p_i(S|e^{\boldsymbol{u}})(r_i - Q(\boldsymbol{u})) - p_i(S|e^{\boldsymbol{u}})p_j(S|e^{\boldsymbol{u}})(r_j - Q(\boldsymbol{u})).
\end{aligned}
$$

Noticing that for $i, j \in S$,

$$
\partial_{u_j} p_i(S|e^{\boldsymbol{u}}) = \begin{cases} -p_j(S|e^{\boldsymbol{u}})p_i(S|e^{\boldsymbol{u}}) & \text{if } j \neq i, \\ (1 - p_i(S|e^{\boldsymbol{u}}))p_i(S|e^{\boldsymbol{u}}) & \text{if } j = i. \end{cases}
$$

Thus

$$
\partial^2_{u_i} Q(\boldsymbol{u}) = p_i(S|e^{\boldsymbol{u}})(1 - 2p_i(S|e^{\boldsymbol{u}}))(r_i - Q(\boldsymbol{u})),
$$

and for $j \neq i$,

$$
\partial_{u_j} \partial_{u_i} Q(\boldsymbol{u}) = -p_i(S|e^{\boldsymbol{u}})p_j(S|e^{\boldsymbol{u}})(r_i + r_j - 2Q(\boldsymbol{u})).
$$

As a result,

$$
\begin{aligned}
|\boldsymbol{w}^\top \nabla^2 Q(\boldsymbol{u})\boldsymbol{w}| &\leq \sum_{i,j \in S} |w_i w_j||\partial_{u_i} \partial_{u_j} Q(\boldsymbol{u})| \\
&\leq \sum_{i \in S}|w_i|^2 p_i(S|e^{\boldsymbol{u}}) + 2\sum_{i \neq j, i, j \in S}|w_i w_j| p_i(S|e^{\boldsymbol{u}})p_j(S|e^{\boldsymbol{u}}) \\
&\leq \sum_{i \in S}|w_i|^2 p_i(S|e^{\boldsymbol{u}}) + \sum_{i \neq j, i, j \in S}(w_i^2 + w_j^2) p_i(S|e^{\boldsymbol{u}})p_j(S|e^{\boldsymbol{u}}) \\
&\leq 3\sum_{i \in S}|w_i|^2 p_i(S|e^{\boldsymbol{u}}) \leq 3\max_{i \in S}|w_i|^2.
\end{aligned}
$$

This finishes the proof. $\qquad\square$

Based on the first and second-order derivatives of $Q$, we now introduce the following perturbation result:

**Proposition 7** (Proposition 3, restated). *For any fixed $\boldsymbol{r}$ and $\boldsymbol{u}, \boldsymbol{u}' \in \mathbb{R}^N$, denote $\boldsymbol{w} := \boldsymbol{u}' - \boldsymbol{u}$, then i) it holds for any $S_0 \in \mathcal{S}$ that*

$$
\begin{aligned}
&|Q(\boldsymbol{u}'; S_0) - Q(\boldsymbol{u}; S_0)| \\
&\leq \sqrt{\sum_{j \in S_0} v_j |r_j - Q(\boldsymbol{u}; S_0)|^2} \cdot \sqrt{\sum_{j \in S_0} p_j(S_0|\boldsymbol{v})p_0(S_0|\boldsymbol{v})w_j^2} + \frac{3}{2}\max_{j \in S_0} w_j^2.
\end{aligned}
$$

*ii) In addition, if $S_0$ is the maximizer of $R(S; e^{\widetilde{\boldsymbol{u}}})$ over $\mathcal{S}$ for some $\widetilde{\boldsymbol{u}} \geq \boldsymbol{u}$ element-wisely, then*

$$
|Q(\boldsymbol{u}'; S) - Q(\boldsymbol{u}; S)| \leq \sqrt{\sum_{j \in S_0} p_j(S_0|e^{\boldsymbol{u}})p_0(S_0|e^{\boldsymbol{u}})w_j^2} + 3\max_{j \in S_0} w_j^2.
$$

*iii) In the uniform-reward setting, where $r_i \equiv 1, \forall i \in [N]$, we have*

$$
|Q(\boldsymbol{u}'; S) - Q(\boldsymbol{u}; S)| \leq \sum_{j \in S_0} p_j(S_0|e^{\boldsymbol{u}})p_0(S_0|e^{\boldsymbol{u}})|w_j| + \frac{3}{2}\max_{j \in S_0} w_j^2
$$

*Proof.* We have by Proposition 6,

$$|Q(\boldsymbol{u}'; S_0) - Q(\boldsymbol{u}; S_0) - \langle \nabla Q(\boldsymbol{u}; S_0), \boldsymbol{w} \rangle|$$

$$= |Q(\boldsymbol{u}') - Q(\boldsymbol{u}; S_0) - \langle \nabla Q(\boldsymbol{u}; S_0), \boldsymbol{w} \rangle| \leq \frac{3}{2} \max_{j \in S_0} w_j^2.$$

On the other hand, we have

$$|\langle \nabla Q(\boldsymbol{u}; S_0), \boldsymbol{w} \rangle| \leq \sum_{j \in S_0} p_j(S_0|\boldsymbol{v})|r_j - Q(\boldsymbol{u}; S_0)||w_j|$$

$$\leq \sqrt{\sum_{j \in S_0} v_j |r_j - Q(\boldsymbol{u}; S_0)|^2} \cdot \sqrt{\sum_{j \in S_0} \frac{v_j w_j^2}{(1 + \sum_{j \in S_0} v_j)^2}}$$

$$\leq \sqrt{\sum_{j \in S_0} v_j |r_j - Q(\boldsymbol{u}; S_0)|^2} \cdot \sqrt{\sum_{j \in S_0} p_j(S_0|\boldsymbol{v})p_0(S_0|\boldsymbol{v})w_j^2}.$$

This finishes the proof of the statement (i).

To prove the second inequality, it suffices to provide upper bound of $\sum_{j \in S_0} v_j |r_j - Q(\boldsymbol{u}; S_0)|$: We first recall the following structural result of the revenue maximizer:

**Proposition 8** (Lemma A.3, [5]). *For any given $\boldsymbol{u}$ denote $\bar{S} := argmax_{S \in \mathcal{S}_K} Q(\boldsymbol{u}; S)$, then it holds that*
*1. $r_i \geq Q(\boldsymbol{u}; \bar{S}), \quad \forall i \in \bar{S}$.*
*2. For any $\boldsymbol{u}' \leq \boldsymbol{u}$ point-wisely, it holds that $Q(\boldsymbol{u}; \bar{S}) \geq Q(\boldsymbol{u}'; \bar{S})$.*

By $S_0$ is the maximizer of $R(S; \widetilde{\boldsymbol{v}})$, we have $r_j \geq Q(\widetilde{\boldsymbol{u}})$ for all $j \in S_0$ by Proposition 8. We further have

$$Q(\boldsymbol{u}; S_0) \leq \max_S R(S; \boldsymbol{v}) \leq \max_S R(S; e^{\widetilde{\boldsymbol{u}}}) = Q(\widetilde{\boldsymbol{u}}),$$

thus then by

$$Q(\boldsymbol{u}; S) = \frac{\sum_{j \in S} r_j v_j}{1 + \sum_{j \in S} v_j} \implies \sum_{j \in S} v_j(r_j - Q(\boldsymbol{u}; S)) = Q(\boldsymbol{u}; S),$$

we have

$$\sum_{j \in S_0} v_j |r_j - Q(\boldsymbol{u}; S_0)|^2 \leq \sum_{j \in S_0} v_j(r_j - Q(\boldsymbol{u}; S_0)) = Q(\boldsymbol{u}; S_0) \leq 1.$$

This finishes the proof of statement (ii).

Finally for statement (iii), we have by

$$r_j - Q(\boldsymbol{u}; S_0) = 1 - \frac{\sum_{j \in S_0} v_j}{1 + \sum_{j \in S_0} v_j} = p_0(S_0|\boldsymbol{v}),$$

$$|\langle \nabla Q(\boldsymbol{u}; S_0), \boldsymbol{w} \rangle| \leq \sum_{j \in S_0} p_j(S_0|\boldsymbol{v})|r_j - Q(\boldsymbol{u}; S_0)||w_j|$$

$$= \sum_{j \in S_0} p_0(S_0|\boldsymbol{v})p_j(S_0|\boldsymbol{v})|w_j|,$$

this finishes the proof. $\qquad\square$

The proof of the above result relies on the condition that $r_j \geq R(S; e^{\boldsymbol{u}})$ for all $j \in S$, which holds only when $S$ is the best assortment under some $\widetilde{\boldsymbol{u}} \geq \boldsymbol{u}$. However, this does not apply to Algorithm 2, which is based on elimination. To handle this, we extend the result to a class of *near-optimal* assortments, which can be applied for analyzing elimination-based algorithms.

**Proposition 9** (Proposition 4 restated). *Under the same notation of Proposition 7 and suppose for some $\widetilde{\boldsymbol{u}} \geq \boldsymbol{u}$ it holds*

$$R(S_0; e^{\widetilde{\boldsymbol{u}}}) \geq \max_{S \in \mathcal{S}} R(S; e^{\widetilde{\boldsymbol{u}}}) - \varepsilon,$$

*then for $\boldsymbol{v} = e^{\boldsymbol{u}}$, $\boldsymbol{w} = \widetilde{\boldsymbol{u}} - \boldsymbol{u}$ and $S^{\star} = argmax_{S \in \mathcal{S}} R(S; \boldsymbol{v})$ we have*

$$R(S^{\star}; \boldsymbol{v}) - R(S_0; \boldsymbol{v}) \leq 2\sqrt{\sum_{j \in S_0} p_j(S_0|\boldsymbol{v}) p_0(S_0|\boldsymbol{v}) w_j^2} + 5 \max_{j \in S_0} w_j^2 + 2\varepsilon.$$

*Proof.* We have by statement (i) of Proposition 7,

$$\begin{aligned}
R(S^{\star}; e^{\boldsymbol{u}}) - R(S_0; e^{\boldsymbol{u}}) &= Q(\boldsymbol{u}; S^{\star}) - Q(\boldsymbol{u}; S_0) \\
&\leq Q(\widetilde{\boldsymbol{u}}; S^{\star}) - Q(\widetilde{\boldsymbol{u}}; S_0) + Q(\widetilde{\boldsymbol{u}}; S_0) - Q(\boldsymbol{u}; S_0) \\
&\leq \varepsilon + |Q(\widetilde{\boldsymbol{u}}; S_0) - Q(\boldsymbol{u}; S_0)| \\
&\leq \varepsilon + \sqrt{\sum_{j \in S_0} v_j(r_j - Q(\boldsymbol{u}; S_0))^2} \sqrt{\sum_{j \in S_0} p_j(S_0|\boldsymbol{v}) p_0(S_0|\boldsymbol{v}) w_j^2} + \frac{3}{2} \max_{j \in S_0} w_j^2.
\end{aligned} \tag{25}$$

Denote

$$S_0^+ := \{j \in S_0 : r_j \geq Q(\boldsymbol{u}; S_0)\}, S_0^- := \{j \in S_0 : r_j < Q(\boldsymbol{u}; S_0)\},$$

then by

$$\sum_{j \in S_0^+} v_j(r_j - Q(\boldsymbol{u}; S_0)) + \sum_{j \in S_0^-} v_j(r_j - Q(\boldsymbol{u}; S_0)) = Q(\boldsymbol{u}; S_0),$$

we have

$$\begin{aligned}
&\sum_{j \in S_0} v_j|r_j - Q(\boldsymbol{u}; S_0)| \\
&= \sum_{j \in S_0^+} v_j(r_j - Q(\boldsymbol{u}; S_0)) - \sum_{j \in S_0^-} (v_j(r_j - Q(\boldsymbol{u}; S_0)) \\
&= 2\sum_{j \in S_0^+} v_j(r_j - Q(\boldsymbol{u}; S_0)) - Q(\boldsymbol{u}; S_0) \\
&\leq 2\sum_{j \in S_0^+} v_j(r_j - Q(\boldsymbol{u}; S^{\star})) + 2\sum_{j \in S_0^+} v_j(Q(\boldsymbol{u}; S^{\star}) - Q(\boldsymbol{u}; S_0)) \\
&\leq 2Q(\boldsymbol{u}; S^{\star}) + 2p_0^{-1}(S_0|\boldsymbol{v})(Q(\boldsymbol{u}; S^{\star}) - Q(\boldsymbol{u}; S_0)),
\end{aligned}$$

where in the last line we have used

$$Q(\boldsymbol{u}; S^{\star}) = argmax_{S \in \mathcal{S}_K} \sum_{j \in S} v_j(r_j - Q(\boldsymbol{u}; S))$$

and $v_j \leq 1 + \sum_{i \in S_0} v_i = p_0^{-1}(S_0|\boldsymbol{v})$.

Denoting $\Delta := (Q(\boldsymbol{u}; S^{\star}) - Q(\boldsymbol{u}; S_0))$, we have then (25) reduces to

$$\begin{aligned}
\Delta &\leq \varepsilon + \sqrt{2 + 2p_0^{-1}(S_0|\boldsymbol{v})\Delta} \cdot \sqrt{\sum_{j \in S_0} p_j(S_0|\boldsymbol{v}) p_0(S_0|\boldsymbol{v}) w_j^2} + \frac{3}{2} \max_{j \in S_0} w_j^2 \\
&\leq \underbrace{\sqrt{\sum_{j \in S_0} p_j(S_0|\boldsymbol{v}) p_0(S_0|\boldsymbol{v}) w_j^2} + \frac{3}{2} \max_{j \in S_0} w_j^2 + \varepsilon}_{:=A} + \underbrace{\sqrt{\sum_{j \in S_0} p_j(S_0|\boldsymbol{v}) w_j^2}}_{:=B} \sqrt{\Delta}.
\end{aligned}$$

Now using the elementary inequality for quadratic equation solutions:

$$\begin{aligned}
x^2 \leq A + Bx &\implies x \leq \frac{B + \sqrt{B^2 + 4A}}{2} \\
&\implies x \leq B + \sqrt{A} \implies x^2 \leq 2B^2 + 2A
\end{aligned}$$

for all $x \geq 0$. We have

$$\Delta \leq 2B^2 + 2A \leq 2\sqrt{\sum_{j \in S_0} p_j(S_0|\boldsymbol{v})p_0(S_0|\boldsymbol{v})w_j^2} + 5\max_{j \in S_0} w_j^2 + 2\varepsilon,$$

as desired. $\qquad\qquad\square$

## D  Proof of Results in Section 4

### D.1  Proof of Proposition 1 and Corollary 1

*Proof.* For each $j \in S^\star$ and any $\lambda > 0$, assume W.L.O.G. that $j$ is exactly contained in $S_1, \ldots, S_{n_j}$, then by

$$H^\star + \lambda I \succeq \sum_{m=1}^{n} \sum_{k \in S_m} p_k(S_m)p_0(S_m)x_k x_k^\top + \lambda I$$

$$\succeq \underbrace{\sum_{m > n_j} \sum_{k \in S_m} p_k(S_m)p_0(S_m)x_k x_k^\top + \lambda I}_{:=Z_j} + \underbrace{\sum_{m \leq n_j} p_j(S_m)p_0(S_m) \, x_j x_j^\top}_{:=\gamma_j},$$

we have then

$$\|x_j\|_{(H^\star + \lambda I)^{-1}}^2 = x_j^\top \left(Z_j + \gamma_j x_j x_j^\top\right)^{-1} x_j$$

$$= x_j^\top Z_j^{-1} x_j \left(1 - \frac{x_j^\top Z_j^{-1} x_j}{1 + \gamma_j x_j^\top Z_j x_j}\right) = \frac{1}{\gamma_j}\left(1 - \frac{1}{\gamma_j x_j^\top Z_j x_j}\right) \leq 1/\gamma_j.$$

Taking limit as $\lambda_j \to 0$ and using the continuity of $\|x_j\|_{(H_D(\theta^\star) + \lambda I)^{-1}}^2$ leads to $\|x_j\|_{H_D^{-1}(\theta^\star)}^2 \leq \gamma_j^{-1}$.

Finally, by

$$\sum_{j \in S^\star} p_j(S^\star)p_0(S^\star)\|x_j\|_{H_D^{-1}(\theta^\star)}^2 \leq \sum_{j \in S^\star} p_j(S^\star)p_0(S^\star)\gamma_j^{-1}$$

$$\leq \sum_{j \in S^\star} \frac{v_j}{(1 + \sum_{k \in S^\star} v_k)^2} \cdot \left[\sum_m \mathbf{1}\{j \in S_m\} \cdot \frac{v_j}{(1 + \sum_{k \in S_m} v_k)^2}\right]^{-1}$$

$$\leq \sum_{j \in S^\star} \frac{v_j}{(1 + \sum_{k \in S^\star} v_k)^2} \cdot \left[\sum_m \mathbf{1}\{j \in S_m\} \cdot \frac{v_j}{(1 + \sum_{k \in S_m} v_k)^2}\right]^{-1}$$

$$\leq \sum_{j \in S^\star} \frac{v_j \cdot \max_{j \in S_m}(1 + \sum_{k \in S_m} v_k)^2}{n_j(1 + \sum_{k \in S^\star} v_k)^2}$$

$$\leq \frac{\max_m(1 + \sum_{k \in S_m} v_k)^2}{\min_j n_j(1 + \sum_{k \in S^\star} v_k)}.$$

This finishes the proof of Corollary 1, and the second inequality implies Proposition 1. $\qquad\square$

### D.2  Details of Proposition 2

In this section we provide the detail on how to plug other confidence region results in Algorithm 1 to achieve a burn-in-free offline learning guarantee. In [44], confidence region results with the radius $\widetilde{O}(\sqrt{d}\|x_j\|_{H_D^{-1}(\theta^\star)})$ are available[3], and we take the result in [44] to establish the confidence region for example. Here we first restate the confidence region bound of [44]:

---

[3]It should be noted that the original result in [44] includes an additional $K$-dependency due to the self-concordant coefficient they established for the MNL likelihood function. This coefficient was later refined by [35], and incorporating their result eliminates the $K$-dependency in [44].

**Lemma 9** (Proposition 3.3 and Lemma C.4 in [44]). *With the selection $\lambda = \sqrt{d/W}$ and denoting*

$$\widehat{\theta}_{D,j}^\lambda := argmin\{x_j^\top \theta : \|\nabla \ell_D^\lambda(\theta) - \nabla \ell_D^\lambda(\widehat{\theta}_D^\lambda)\|_{(H_D(\theta)+\lambda I)^{-1}} \le 4\sqrt{d(1+W)}\log(4Knd/\delta)\}, \tag{26}$$

*it holds that with probability at least $1 - \delta$,*

$$\|\widehat{\theta}_{D,j}^\lambda - \theta^\star\|_{H_D(\theta^\star)+\lambda I} \le 4\sqrt{d(1+W)^3}\log(4Knd/\delta).$$

With Lemma 9 and (26), we now assign the LCB values for each $j \in [N]$ as

$$\widetilde{u}_j^{\mathrm{LCB}} := x_j^\top \widehat{\theta}_{D,j}^\lambda, \quad \widetilde{v}_j^{\mathrm{LCB}} = \exp(\widetilde{u}_j^{\mathrm{LCB}}) \text{ if } j \in \cup_{k=1}^n S_k, \quad \widetilde{v}_j^{\mathrm{LCB}} = 0 \text{ otherwise.}$$

It can be seen from Lemma 9 that with probability at least $1 - \delta$,

$$0 \le u_j^\star - \widetilde{u}_j^{\mathrm{LCB}} \lesssim \sqrt{d(1+W)^3}\log(4Knd/\delta)\|x_j\|_{H_D(\theta^\star)+\lambda I}.$$

Now replacing the term $u^\star - \widehat{u}_j$ by $u_j^\star - \widetilde{u}_j^{\mathrm{LCB}}$ the right-hand-side upper bound our analysis in Section A.3 leads to Proposition 2, as desired.

# E  Proof of Results in Section 5

## E.1  Proof of Lemma 1

Lemma 1 is a direct corollary of the following result from [47, 53]:

**Lemma 10.** *For any given $\mathcal{X} \subset \{x \in \mathbb{R}^d : \|x\|_2 \le 1\}$ threshold $\mu > 0$ and regularizer $\lambda > 0$, the following procedure starting with $\mathcal{C} = \emptyset$:*

> **while** $\exists x \in \mathcal{X}$ so that $\|x\|_{(V_{\mathcal{C}}^\lambda)^{-1}}^2 > \mu$ for $V_{\mathcal{C}}^\lambda := \sum_{z \in \mathcal{C}} zz^\top + \lambda I$ **do:** *add this $x$ to $\mathcal{C}$.*

*will stop in at most*

$$C_{\max} := \frac{e}{e-1}\frac{1+\mu}{\mu}d\left(\log\left(1+\frac{1}{\mu}\right) + \log\left(1+\frac{1}{\lambda}\right)\right).$$

*steps.*

Applying this lemma with

$$\lambda = 1, \mu = \left(\frac{1}{64\sqrt{d\log(NT)}} \wedge \frac{1}{64W}\right)^2$$

to each $\ell$ separately then leads to the desired result.

## E.2  Proof of Theorem 4

Based on the burn-in condition established in Lemma 1, which incurs at most poly-logarithmic regret in $T$, it remains to bound the regret incurred from $\tau + 1$ to $T$.

Again, it suffices to perform regret analysis under the inequalities (9) and (10), under which we can show that

$$2^{-8} \le \frac{\widetilde{W}_{t,S}^\ell}{\sqrt{\sum_{j \in S} p_j(S|\boldsymbol{v})p_0(S|\boldsymbol{v})} \cdot \sqrt{\sum_{j \in S} p_j(S|\boldsymbol{v})p_0(S|\boldsymbol{v})\|x_j\|_{H_{t,\ell}^{-1}}^2}} \le 2^8$$

holds using the same argument for establishing (11).

Now noticing that with the uniform revenue assumption $r_i \equiv \bar{r}$ for some $0 \le \bar{r} \le 1$, statement i) in Proposition 3 can be refined to

$$|R(S_0|\boldsymbol{v}') - R(S_0|\boldsymbol{v})| \le \sqrt{\sum_{j \in S_0} v_j|r_j - R(S_0|\boldsymbol{v})|^2} \cdot \sqrt{\sum_{j \in S_0} p_j(S_0|\boldsymbol{v})p_0(S_0|\boldsymbol{v})w_j^2} + \frac{3}{2}\max_{j \in S_0} w_j^2$$

$$\le \sqrt{\sum_{j \in S_0} p_0(S_0|\boldsymbol{v})p_j(S_0|\boldsymbol{v})} \cdot \sqrt{\sum_{j \in S_0} p_j(S_0|\boldsymbol{v})p_0(S_0|\boldsymbol{v})w_j^2} + \frac{3}{2}\max_{j \in S_0} w_j^2,$$

where the last inequality is by

$$\sum_{j \in S_0} v_j |r_j - R(S_0|\boldsymbol{v})|^2 = \sum_{j \in S_0} \bar{r}^2 v_j \left(1 - \frac{\sum_{j \in S_0} v_j}{1 + \sum_{j \in S_0} v_j}\right)^2 = \frac{\bar{r}^2 \sum_{j \in S_0} v_j}{(1 + \sum_{j \in S_0} v_j)^2} \leq \sum_{j \in S_0} p_0(S_0|\boldsymbol{v}) p_j(S_0|\boldsymbol{v}).$$

With this result, we have the following analogue of Lemma 2 directly in this setting: For every $\ell$ and $S \in \mathcal{A}_\ell$ it holds under (9) and (10) that

$$|R_{t,\ell}^{\text{UCB}}(S) - R(S|\theta^\star)| \leq 2^8 \widetilde{W}_{t,S}^\ell. \tag{27}$$

Plugging (27) to the subsequent analysis in the proof of Theorem 3 then leads to

$$\sum_{t \in \mathcal{T}_{a,\ell}} R(S^\star) - R(S_t) \lesssim \sum_{t \in \mathcal{T}_{a,\ell}} \sqrt{\sum_{j \in S_t} p_0(S_t|\boldsymbol{v}) p_j(S_t|\boldsymbol{v})} \cdot \sqrt{\sum_{j \in S_t} p_j(S_t|\boldsymbol{v}) p_0(S_t|\boldsymbol{v}) (w_{tj}^\ell)^2} + \frac{3}{2} \max_{j \in S_t}(w_{tj}^\ell)^2,$$

$$\lesssim \sqrt{\sum_{t \in \mathcal{T}_{a,\ell}} \sum_{j \in S_t} p_0(S_t|\boldsymbol{v}) p_j(S_t|\boldsymbol{v})} \cdot \sqrt{\sum_{t \in \mathcal{T}_{a,\ell}} \sum_{j \in S_t} p_j(S_t|\boldsymbol{v}) p_0(S_t|\boldsymbol{v}) (w_{tj}^\ell)^2} + \sum_{t \in \mathcal{T}_{a,\ell}} \max_{j \in S_t}(w_{tj}^\ell)^2$$

$$\lesssim \sqrt{\sum_{t \in \mathcal{T}_{a,\ell}} \sum_{j \in S_t} p_0(S_t|\boldsymbol{v}) p_j(S_t|\boldsymbol{v}) \cdot (d + W) \log(NT) + \frac{d \log(NT)}{\kappa}}$$

$$\lesssim \sqrt{\left(\kappa^\star T + \sum_{t \in \mathcal{T}_{a,\ell}} R(S^\star) - R(S_t)\right) \cdot (d + W) \log(NT) + \frac{d \log(NT)}{\kappa}},$$

where the last line is by Lemma 11 of [44]. Now if we denote $\Delta := \sum_{t \in \mathcal{T}_{a,\ell}} R(S^\star) - R(S_t)$ and apply the fact

$$\Delta \lesssim A\sqrt{\Delta} + B \implies \Delta \lesssim A^2 + B$$

with $A = (d + W)\log(NT)$ and $B = \sqrt{\kappa^\star(d+W)T\log(NT)} + \frac{d\log(NT)}{\kappa}$, the desired result holds.

### E.3 Proof of Theorem 5

In this section, we show the following problem-dependent lower bound result:

**Theorem 5** (Problem-Dependent Lower Bound). *For any given $0 < \bar{\kappa} < 1$, we can find a class of problem instances $\mathcal{V}$ with uniform revenue and $d$-dimensional linear MNL choice feedback so that for any instance in $\mathcal{V}$ the corresponding parameter $\kappa^\star \leq \bar{\kappa}$ and for any policy $\pi$, there exists some instance in $\mathcal{V}$ so that*

$$Reg(T) \gtrsim \sqrt{\bar{\kappa}dT}$$

*over such instance.*

The construction of hard instance for proving Theorem 5 relies only on a minor modification of the proof in [17], and we just provide the detailed proof here for completeness.

**Step 1: Construction of Hard Instances.** We let $r_1 = \cdots = r_N = 1$ for all instances constructed later, and given any $K$-sized assortment $S$, we consider the corresponding attraction value construction as

$$v^S \doteq \begin{cases} \frac{\kappa^\star}{K}(1 + \varepsilon) & \text{if } i \in S, \\ \frac{\kappa^\star}{K} & \text{otherwise.} \end{cases}$$

And we define the problem instance set as

$$\mathcal{V}_K \doteq \{v^S : S \in \mathcal{S}_K\}.$$

It can be seen that we always have $S^\star(v^S) = S$ for every $v$, and

$$\frac{\kappa^\star}{9} \leq \sum_{i \in S^\star} p_i(S^\star) p_0(S^\star) = \frac{\kappa^\star(1 + \varepsilon)}{(1 + \kappa^\star(1 + \varepsilon))^2} \leq 2\kappa^\star$$

holds for every $v^S$ by definition.

Since there exists a one-to-one correspondence between $\mathcal{S}_K$ and $\mathcal{V}_K$ via $S \to v^S$, we interchangeably use the notations $v^S$ and $S$ for convenience.

**Step 2: Verifying the Separation Condition.** We have the following separation result over $\mathcal{S}$

**Proposition 10.** *For every fixed $S_0 \in \mathcal{S}$ and corresponding $v^{S_0} \in \mathcal{V}_K$, it holds for any $S \in \mathcal{S}$ that*

$$\mathrm{SubOpt}(S|v^{S_0}) \geq \frac{(K - |S \cap S_0|)\kappa^\star \varepsilon}{9K},$$

*as desired.*

*Proof of Proposition 10.* Denoting $R(\cdot|v^{S_0})$ by $R(\cdot)$ thorough the proof for simplicity, we have

$$R(S_0) = \frac{\kappa^\star(1+\varepsilon)}{1 + \kappa^\star(1+\varepsilon)}$$

and for any $S$,

$$R(S) = \frac{\kappa^\star(1 + |S \cap S_0|\varepsilon/K)}{1 + \kappa^\star(1 + \underbrace{|S \cap S_0|\varepsilon/K}_{\doteq \varepsilon'})},$$

thus

$$
\begin{aligned}
R(S_0) - R(S) &= \frac{\kappa^\star(1+\varepsilon)(1 + \kappa^\star(1+\varepsilon')) - \kappa^\star(1+\varepsilon')(1 + \kappa^\star(1+\varepsilon))}{(1 + \kappa^\star(1+\varepsilon))(1 + \kappa^\star(1+\varepsilon'))} \\
&= \frac{\kappa^\star(\varepsilon - \varepsilon')}{(1 + \kappa^\star(1+\varepsilon))(1 + \kappa^\star(1+\varepsilon'))} \geq \frac{\kappa^\star(\varepsilon - \varepsilon')}{9} \\
&= \frac{\kappa^\star(K - |S \cap S_0|)\varepsilon}{9K}
\end{aligned}
$$

$\square$

**Step 3: Decomposing the Regret.** Based on Proposition 10, we have

$$
\max_{\boldsymbol{v} \in \mathcal{V}_K} \mathrm{Reg}_\pi(T; \boldsymbol{v}) \geq \frac{1}{|\mathcal{S}_K|} \sum_{\boldsymbol{v} \in \mathcal{V}_K} \sum_{t=1}^T \mathbb{E}_{\boldsymbol{v}}[\mathrm{SubOpt}(S_t; \boldsymbol{v})]
$$

$$
\geq \frac{\kappa^\star \varepsilon}{9K|\mathcal{S}_K|} \sum_{S \in \mathcal{S}_K} \sum_{t=1}^T \mathbb{E}_S[K - |S_t \cap S^\star(\boldsymbol{v})|]
$$

$$
= \frac{\kappa^\star \varepsilon}{9|\mathcal{S}_K|} \sum_{S \in \mathcal{S}_K} \left(T - \frac{1}{K} \sum_{i \in S} \sum_{t=1}^T \mathbb{E}_S[\mathbf{1}\{i \in S_t\}]\right)
$$

$$
= \frac{\kappa^\star \varepsilon}{9} \left(T - \frac{1}{K|\mathcal{S}_K|} \sum_{S \in \mathcal{S}_K} \sum_{i \in S} \mathbb{E}_S[N_i]\right).
$$

Now for each $i$, we define $\mathcal{S}_{K-1}^{(i)} \doteq \mathcal{S}_{K-1} \cap \{S \subset [N] : i \notin S\}$, then it holds that

$$
\sum_{S \in \mathcal{S}_K} \sum_{i \in S} \mathbb{E}_S[N_i] = \sum_{i=1}^N \sum_{S \in \mathcal{S}_K : i \in S} \mathbb{E}_S[N_i] = \sum_{i=1}^N \sum_{S' \in \mathcal{S}_{K-1}^{(i)}} \mathbb{E}_{S' \cup \{i\}}[N_i]
$$

$$
= \sum_{i=1}^N \sum_{S' \in \mathcal{S}_{K-1}^{(i)}} \left(\mathbb{E}_{S' \cup \{i\}}[N_i] - \mathbb{E}_{S'}[N_i]\right) + \sum_{i=1}^N \sum_{S' \in \mathcal{S}_{K-1}^{(i)}} \mathbb{E}_{S'}[N_i].
$$

Now for the second term, we have

$$\sum_{i=1}^{N} \sum_{S' \in \mathcal{S}_{K-1}^{(i)}} \mathbb{E}_{S'}[N_i] = \sum_{S' \in \mathcal{S}_{K-1}} \sum_{i \notin S'} \mathbb{E}_{S'}[N_i] \leq KT|\mathcal{S}_{K-1}|.$$

For the first term, by Pinsker's inequality, denoting the probability induced by environment $S' \cup \{i\}$ as $P$ and induced by $S'$ as $Q$, then

$$|\mathbb{E}_{S' \cup \{i\}}[N_i] - \mathbb{E}_{S'}[N_i]| \leq \sum_{t=1}^{T} t|P_{S'}(N_i = t) - Q_{S'}(N_i = t)|$$

$$\leq T\sqrt{\frac{1}{2}\mathrm{KL}(P\|Q)}.$$

As a result, we have

$$\max_{\boldsymbol{v} \in \mathcal{V}_K} \mathrm{Reg}_\pi(T; \boldsymbol{v}) \geq \frac{\kappa^\star \varepsilon}{9}\Big(T - \frac{1}{K|\mathcal{S}_K|} \sum_{S \in \mathcal{S}_K} \sum_{i \in S} \mathbb{E}_S[N_i]\Big)$$

$$\geq \frac{\kappa^\star \varepsilon}{9}\Big(T - \frac{T|\mathcal{S}_{K-1}|}{|\mathcal{S}_K|} - \frac{T}{\sqrt{2}K|\mathcal{S}_K|} \sum_{i=1}^{N} \sum_{S' \in \mathcal{S}_{K-1}^{(i)}} \sqrt{\mathrm{KL}(P_{S'}\|Q_{S'})}\Big)$$

$$\geq \frac{\kappa^\star \varepsilon T}{9}\Big(\frac{2}{3} - \frac{1}{K|\mathcal{S}_K|} \sum_{i=1}^{N} \sum_{S' \in \mathcal{S}_{K-1}^{(i)}} \sqrt{\frac{1}{2}\mathrm{KL}(P_{S'}\|Q_{S'})}\Big).$$

Where the last line is by

$$\frac{|\mathcal{S}_{K-1}|}{|\mathcal{S}_K|} = \frac{\binom{N}{K-1}}{\binom{N}{K}} = \frac{K}{N - K + 1} \leq \frac{1}{3}$$

when $4K \leq N$.

**Step 4: Upper Bounding the KL Divergence.** We have for every $S' \in \mathcal{S}_{K-1}^{(i)}$, it holds that

$$\mathrm{KL}(P_{S'}\|Q_{S'}) \leq \sum_{\bar{S} \in \mathcal{S}} \mathbb{E}_{S'}[N(\bar{S})]\mathrm{KL}\big(P_{S'}(\cdot|\bar{S})\|Q_{S'}(\cdot|\bar{S})\big)$$

$$\leq \sum_{\bar{S} \in \mathcal{S}: i \in \bar{S}} \mathbb{E}_{S'}[N(\bar{S})]\mathrm{KL}\big(P_{S'}(\cdot|\bar{S})\|Q_{S'}(\cdot|\bar{S})\big)$$

Where the last line is by $\mathrm{KL}\big(P_{S'}(\cdot|\bar{S})\|Q_{S'}(\cdot|\bar{S})\big) = 0$ as long as $i \notin \bar{S}$.

For every $\bar{S}$ containing $i$, we have denote $K' = |\bar{S}|$ and $p_j \doteq P_{S'}(j|\bar{S}), q_j \doteq Q_{S'}(j|\bar{S})$ for $j \in \bar{S}_+$, then
i) For $j = 0$,

$$|p_0 - q_0| = \frac{K}{K + \kappa^\star\big(K' + |\bar{S} \cap S'|\varepsilon\big)} - \frac{K}{K + \kappa^\star\big(K' + (|\bar{S} \cap S'| + 1)\varepsilon\big)}$$

$$\leq \frac{K\kappa^\star}{\big[K + \kappa^\star\big(K' + |\bar{S} \cap S'|\varepsilon\big)\big]^2} \leq \frac{\kappa^\star \varepsilon}{K}.$$

ii) For $j \neq i$,

$$|p_j - q_j| \leq \frac{\kappa^\star(1 + \varepsilon)}{K}|p_0 - q_0| \leq \frac{2(\kappa^\star)^2 \varepsilon}{K^2}.$$

iii) For $j = i$,

$$|p_i - q_i| = \frac{\kappa^\star(1 + \varepsilon)}{K + \kappa^\star\big(K' + (|\bar{S} \cap S'| + 1)\varepsilon\big)} - \frac{\kappa^\star}{K + \kappa^\star\big(K' + |\bar{S} \cap S'|\varepsilon\big)}$$

$$\leq \frac{\varepsilon\kappa^\star(K + \kappa^\star(K' + |\bar{S} \cap S'|\varepsilon)) - \kappa^\star \varepsilon}{K^2} \leq \frac{2\varepsilon\kappa^\star}{K},$$

Now applying the following Proposition from [17]:

**Proposition 11.** *Let $P$ and $Q$ be two categorical distributions on $J$ items, with parameters $p_1, \cdots, p_J$ and $q_1, \cdots, q_J$ respectively. Denote also $\varepsilon_j \doteq p_j - q_j$. Then $\mathrm{KL}(P\|Q) \leq \sum_{j=1}^{J} \varepsilon_j^2/q_j$.*

we get

$$
\sum_{\bar{S} \in \mathcal{S}: i \in \bar{S}} \mathbb{E}_{S'}[N(\bar{S})]\mathrm{KL}\big(P_{S'}(\cdot|\bar{S}) \| Q_{S'}(\cdot|\bar{S})\big)
$$

$$
\leq \mathbb{E}_{S'}[N_i]\Big(\frac{\kappa^2 \varepsilon^2}{K} + \frac{2K'(\kappa^\star)^4 \varepsilon^2}{\kappa^\star K^4} + \frac{4\varepsilon^2 (\kappa^\star)^2}{\kappa^\star K}\Big) \leq \mathbb{E}_{S'}[N_i]\frac{24\varepsilon^2 \kappa^\star}{K}.
$$

**Final Step: Putting All Together.** Using the above upper bound of KL divergence, we get

$$
\frac{\kappa^\star \varepsilon T}{9}\Big(\frac{2}{3} - \frac{1}{K|\mathcal{S}_K|} \sum_{i=1}^{N} \sum_{S' \in \mathcal{S}_{K-1}^{(i)}} \sqrt{\frac{1}{2}\mathrm{KL}(P_{S'}\|Q_{S'})}\Big)
$$

$$
\geq \frac{\kappa^\star \varepsilon T}{9}\Big(\frac{2}{3} - \frac{1}{K|\mathcal{S}_K|} \sum_{i=1}^{N} \sum_{S' \in \mathcal{S}_{K-1}^{(i)}} \sqrt{\frac{1}{2}\mathbb{E}_{S'}[N_i]\frac{24\varepsilon^2 \kappa^\star}{K}}\Big)
$$

$$
\geq \frac{\kappa^\star \varepsilon T}{9}\Big(\frac{2}{3} - \frac{1}{K|\mathcal{S}_K|} \sum_{S' \in \mathcal{S}_{K-1}} \sum_{i \notin S'} \sqrt{\frac{1}{2}\mathbb{E}_{S'}[N_i]\frac{24\varepsilon^2 \kappa^\star}{K}}\Big)
$$

$$
\geq \frac{\kappa^\star \varepsilon T}{9}\Big(\frac{2}{3} - \frac{1}{K|\mathcal{S}_K|} \sum_{S' \in \mathcal{S}_{K-1}} \sqrt{N} \sqrt{\sum_{i \notin S'} \frac{1}{2}\mathbb{E}_{S'}[N_i]\frac{24\varepsilon^2 \kappa^\star}{K}}\Big)
$$

$$
\geq \frac{\kappa^\star \varepsilon T}{9}\Big(\frac{2}{3} - \frac{|\mathcal{S}_{K-1}|}{K|\mathcal{S}_K|} \sqrt{12NT\varepsilon^2 \kappa^\star}\Big)
$$

$$
\geq \frac{\kappa^\star \varepsilon T}{9}\Big(\frac{2}{3} - \frac{1}{N-K+1} \sqrt{12NT\varepsilon^2 \kappa^\star}\Big) \geq \frac{\kappa^\star \varepsilon T}{9}\Big(\frac{2}{3} - \sqrt{48T\varepsilon^2 \kappa^\star/N}\Big).
$$

Selecting $\varepsilon^2 = \frac{1}{512}\sqrt{\frac{N}{T\kappa^\star}}$ then leads to the $\Omega(\sqrt{\kappa^\star NT})$ lower bound, as desired.

### E.4 Extension to Time-Varying Contexts

Another problem setting widely adopted in linear MNL and generalized linear bandit frameworks is the adversarial context setting with an initial exploration period [38, 18, 31, 42]. In this setting, instead of always using the same feature map $x_j$, we allow the feature vector $x_{t,j}$ to vary with $t$. Furthermore, during the initial exploration period, the observed features $x_{t,j}$ are drawn i.i.d. from some distribution $P_0$ with $\lambda_{\min}(\mathbb{E}_{x \sim P_0}[xx^\top]) \geq \sigma_0$ for some $\sigma_0 > 0$. In general, The fixed action setting does not fit this framework as it violates the eigenvalue lower bound assumption, making previous exploration-phase designs inapplicable. While our algorithmic description focuses on the fixed feature set setting, it can be readily extended to the time-varying context setting, achieving a regret of $\widetilde{O}(\sqrt{dT} + \kappa^{-1}d^2 K)$. This holds under the same stochastic context assumption during the initial exploration phase and allows for an even simpler design in the exploration phase, thanks to the generality of Theorem 1.

In this section, we extend Algorithm 2 to the setting that the observed context at each round can be different vectors in $\mathbb{R}^d$, more precisely, at each time $t$, the $i$-th item is associated with a feature map $x_{t,i} \in \mathbb{R}^d$. And we impose the following eigenvalue assumption in the exploration phase, as in [42, 18]:

**Assumption 2.** *For some given $t_0$, $\{x_{t,i}\}_{i \in [N], t \in [t_0]}$ are generated i.i.d. from some unknown distribution $P$ supported on the $d$-dimensional unit ball, with $\lambda_{\min}(\mathbb{E}_{x \sim P}[xx^\top]) \geq \sigma_0$ for some $\sigma_0 > 0$.*

We first present the modified algorithm for time-varying contexts in Algorithm 3, with modifications highlighted in blue. In the elimination phase, the only change is that item-wise uncertainty levels are computed over $x_{t,i}$ instead of $x_i$ for each $i$, which then affects $W_{t,S}^\ell$ and the UCB revenues. Thus the same analysis as in Section A.3 can be conducted to derive the same regret bound once the burn-in condition can be verified.

The main difference lies in the initial exploration phase, where any $K$-sized assortment can be selected for efficient exploration due to Assumption 2.

---

**Algorithm 3** SupLinearMNL with Time-Varying Contexts

---

1: **Input:** Time horizon $T$, exploration length $n_0, \tau$.
   **initialize** $S = \log T, , \Psi_1 = \cdots = \Psi_{S+1} = \emptyset$.
2: **for** $t = 1, \ldots, (S+1)\tau$ **do**
3:     Select arbitrary assortment in $\mathcal{S}_K$, add $t$ into $\Psi_{\lceil t/\tau \rceil}$
4: **end for**
5: $\Psi_0 \leftarrow \emptyset$, compute $\widehat{\theta}_0$ as in (5).
6: **for** $t = (S+1)\tau + 1, \ldots, T$ **do**
7:     set $\mathcal{A}_1 = \mathcal{S}_K, S_t = \emptyset, \ell = 1$
8:     **while** $S_t = \emptyset$ **do**
9:         Compute $W_{t,S}^\ell, R_{t,\ell}^{\text{UCB}}(S), \forall S \in \mathcal{A}_\ell$ as in (7), (8) , with $w_{t,i}^\ell := 72\|x_{t,i}\|_{H_{t,\ell}^{-1}(\widehat{\theta}_0)}\sqrt{\log(NT)}$
10:         **if** $W_{t,S}^\ell > 2^{-\ell}$ for some $S \in \mathcal{A}_\ell$ **then**
11:             select such $S \in \mathcal{A}_\ell$.
12:             $\Psi_\ell \leftarrow \Psi_\ell \cup \{t\}$
13:         **else if** $W_{t,S}^\ell \le 1/\sqrt{T}$ for all $S \in \mathcal{A}_\ell$ **then**
14:             take the action $S_t = \arg\max_{S \in \mathcal{A}_\ell} R_{t,\ell}^{\text{UCB}}(S)$
15:             $\Psi_0 \leftarrow \Psi_0 \cup \{t\}$
16:         **else**
17:             $\widehat{R} \leftarrow \max_{S \in \mathcal{A}_\ell} R_{t,\ell}^{\text{UCB}}(S)$
18:             $\mathcal{A}_{\ell+1} \leftarrow \left\{ S \in \mathcal{A}_\ell, R_{t,\ell}^{\text{UCB}}(S) \ge \widehat{R} - 2^{-\ell+2} \right\}$
19:             $\ell \leftarrow \ell + 1$
20:         **end if**
21:     **end while**
22: **end for**

---

Now it sufficient to prove the following burn-in condition guarantee:

**Lemma 11.** *With the selection* $\tau = \Omega(\sigma_0^{-1}[d\log T/\sigma_0 + \sqrt{d\log(NT)} \vee K^{-1}\sqrt{W})$, *it holds that with probability at least* $1 - 2/T$, *the event* $\widetilde{\mathcal{E}}_1 \cap \widetilde{\mathcal{E}}_2$, *with*

$$\widetilde{\mathcal{E}}_1 := \{\frac{1}{2}H_{t,\ell}(\theta^\star) \preceq H_{t,\ell}(\widehat{\theta}_0) \preceq 2H_{t,\ell}(\theta^\star), \quad \forall t > \tau, \ell \in [S]\},$$

$$\widetilde{\mathcal{E}}_2 := \{|x_{t,j}^\top(\widehat{\theta}_{t,\ell}^\lambda - \theta^\star)| \le 72\sqrt{\log(NT)}\|x_{t,j}\|_{(H_{t,\ell}^\lambda(\widehat{\theta}_0))^{-1}}, \quad \forall t > \tau, j \in [N], \ell \in [S]\},$$

*holds.*

*Proof.* We need only show that with probability at least $1 - 1/T$, after the exploration phase, it holds for every $\ell$ that

$$\lambda_{\min}(H_{t,\ell}(\theta^\star)) \ge 64K\sqrt{d\log(NT)} \vee 64\sqrt{W}, \tag{28}$$

From this, we obtain

$$\|x\|_{H_{t,\ell}(\theta^\star)} \lesssim \frac{1}{64K\sqrt{d\log(NT)}} \wedge \frac{1}{64\sqrt{W}}, \quad \forall \|x\|_2 \le 1,$$

which then allows the result to follow naturally from Theorem 1 and Proposition 5.

Now to prove (28), we simply recall the following result in [42] and [38]:

**Proposition 12** (Proposition 1 in [42]& Proposition 1 in [38])**.** *For any constant $B > 0$, there exists some absolute constant $c_1, c_2 > 0$ so that with the selection*

$$\kappa\tau \ge \frac{1}{K}\left(\frac{c_1\sqrt{d} + c_2\sqrt{2\log T}}{\sigma_0}\right)^2 + \frac{2B}{K\sigma_0},$$

*it holds with probability at least $1 - 1/T^2$ that $H_{t,\ell}(\theta^\star) \succeq BI$.*

Now selecting $\tau$ as in Proposition 12 with $B = 64K\sqrt{d\log(NT)} \vee 64\sqrt{W}$ then finishes our proof. $\square$

### E.5 Extension of OFU-MNL [44]

In this section, we show that a simple plug-in argument based on the developed perturbation result Proposition 3 can extend the result in Theorem 3.4 of [44] for uniform revenue setting to the general revenue setting:

**Theorem 6.** *The OFU-MNL algorithm (Algorithm 2, [44]) satisfies the regret*

$$Reg(T) \lesssim d\sqrt{T} + \kappa^{-1}d^2$$

*up to logarithmic factors.*

*Proof of Theorem 6.* Simply noticing that the OFU-MNL algorithm always taking the optimistic action

$$S_t := \mathrm{argmax}_{S \in \mathcal{S}_K} R(S|\boldsymbol{v}_t^{\mathrm{UCB}})$$

for

$$v_{ti}^{\mathrm{UCB}} := \max_{\theta \in \Theta} \exp(x_{ti}^\top \theta) \quad \forall i \in [N],$$

with $\mathcal{C}_t$ specified as in Lemma 9.

Proposition 3.3 in [44] has shown that $\boldsymbol{v}_t^{\mathrm{UCB}} \geq \boldsymbol{v}, \forall t \in [T]$ with probability at least $1 - O(1/T)$, thus then we have under such event, it holds that $r_j \geq R(S_t|\boldsymbol{v}_t^{\mathrm{UCB}}) \geq R(S^\star|\boldsymbol{v})$. Now denote $w_{tj} := v_{tj}^{\mathrm{UCB}} - v_{tj}$, we have then

$$\mathrm{Reg}(T) = \sum_{t=1}^{T} R(S^\star|\boldsymbol{v}) - R(S_t|\boldsymbol{v}) \leq \sum_{t=1}^{T} R(S^\star|\boldsymbol{v}_t^{\mathrm{UCB}}) - R(S_t|\boldsymbol{v})$$

$$\leq \sum_{t=1}^{T} R(S_t|\boldsymbol{v}_t^{\mathrm{UCB}}) - R(S_t|\boldsymbol{v}) \leq_{\text{(i)}} \sqrt{\sum_{j \in S_t} p_j(S_t|\boldsymbol{v})p_0(S_t|\boldsymbol{v})w_{tj}^2} + 3\max_{j \in S_t} w_{tj}^2$$

$$\lesssim_{\text{(ii)}} \sqrt{d \sum_{j \in S_t} p_j(S_t|\boldsymbol{v})p_0(S_t|\boldsymbol{v})\|x_{tj}\|_{H_t^{-1}(\theta^\star)}^2} + 3d\max_{j \in S_t}\|x_{tj}\|_{H_t^{-1}(\theta^\star)}^2.$$

where (i) is by statement ii) of Proposition 3, (ii) is by Proposition 3.3 and Lemma 9. Finally, applying the elliptic potential lemma presented in Lemma 5 leads to the desired result. $\square$

## F Experiment Results

In this section, we present additional numerical results to illustrate the effect of the burn-in condition and to compare Algorithm 1 with existing offline assortment optimization benchmarks.

### F.1 Sensitivity to Burn-in Condition

To better understand the trade-off between the tightness of our confidence intervals (CIs) and the burn-in condition, we conduct a set of numerical experiments comparing our bound in Theorem 2 with that of [44]. Our CIs are theoretically tighter than those in [44, 35, 3] in that the confidence radius is smaller by a $\sqrt{d}$ factor, but requires an additional burn-in condition and a conditional independence assumption. The following simulation demonstrates how violating the burn-in condition may lead to the failure of our CI guarantee, while satisfying it yields improved tightness.

#### F.1.1 Evaluation and Results

We compare the empirical tightness of the two confidence bounds using the following ratio:

$$\text{CI Ratio} := \max_i \frac{|\widehat{\theta}_i - \theta_i^\star|}{\text{CI at } e_i},$$

where the denominator corresponds to the theoretical confidence radius at coordinate $e_i$. A smaller CI ratio indicates a tighter alignment between the theoretical CI and the actual estimation error, and thus a lower likelihood of CI violation.

We evaluate the CI ratio under different burn-in constants, defined as

$$\zeta := \max_{k \leq n,\, j \in S_k} \|x_{kj}\|_{H^{-1}(\theta^\star)}.$$

Theoretically, our CI guarantee in Theorem 2 requires $\zeta \lesssim 1/\sqrt{d}$, whereas the bound in [44] imposes no such restriction.

| Value of $\zeta$ | 0.130 | 0.134 | 0.141 | 0.160 | 0.335 |
|---|---|---|---|---|---|
| CI ratio (Theorem 1) | 2.563 | 2.831 | 3.147 | 4.052 | 5.617 |
| CI ratio ([44]) | 3.495 | 3.480 | 3.420 | 3.270 | 2.852 |

Table 2: Comparison of CI ratios under varying burn-in constants.

We present the result in Table 2, and defer the detailed experiment setup in the next section. As $\zeta$ increases, our CI ratio grows noticeably, reflecting a higher risk of CI violation when the burn-in condition is not met. In contrast, the ratio for [42] remains stable across $\zeta$, consistent with its theory-independent nature. This comparison highlights a fundamental trade-off: our CI achieves tighter bounds under well-conditioned data but is more sensitive to initialization.

### F.1.2 Experimental Setup

We describe below how the feature map, parameter, and assortment sets are constructed to obtain results in Table 2.

**Feature Set and Parameters.** For given $(d, K)$ (assuming $d$ even) and a parameter radius $W > 0$, we define the item features as follows:

- **Type-1 items:** For $i \leq d - 1$, $x_i = e_i$ (canonical basis vectors).
- **Type-2 item:** $x_d = (1, 1, \ldots, 1)/\sqrt{d}$.

The true parameter is set as $\theta^\star = (W, W, \ldots, W)/\sqrt{d}$.

**Assortment Sets.** We consider two types of assortments:

- **Type-1 assortment:** $S_0 = \{d\}$, containing only the Type-2 item.
- **Type-2 assortments:** Let $m = \lceil d/K \rceil$. For each $1 \leq k \leq m$, define

$$S_k = \{(m - 1)K + 1, (m - 1)K + 2, \ldots, \min(mK, d)\}.$$

**Observed Data.** Given integers $n_1, n_2$, we generate $n_1$ copies of $S_1, \ldots, S_m$ and $n_2$ copies of $S_0$, with choices sampled under the linear MNL model parameterized by $(X, \theta^\star)$ as above.

**Relation Between $W$ and Burn-in Constant.** Under this construction, $\zeta$ increases monotonically with $W$. The mapping between $W$ and $\zeta$ used in our experiments is shown below.

| Value of $W$ | 0.5 | 0.841 | 1.41 | 2.37 | 4.0 |
|---|---|---|---|---|---|
| Value of $\zeta$ | 0.130 | 0.134 | 0.141 | 0.160 | 0.335 |

Table 3: Mapping between $W$ and burn-in constant $\zeta$.

All results reported in Table 2 are based on $d = 6$, $K = 3$, and $n_1 = n_2 = 500$. This setup allows us to systematically vary $W$ and thus $\zeta$, thereby illustrating the relationship between burn-in strength and confidence interval validity.

## F.2 Results for Offline Assortment Optimization

Since in online setting, Algorithm 2 is computationally inefficient. We focus on Algorithm 1 in the offline setting. We consider two variants of our approach based on different confidence intervals: one derived from Theorem 2 and the other from Lemma 9, denoted by **LCB-MLE** and **LCB-MLE-v2**, respectively. These experiments are designed to examine the trade-off between statistical conservativeness and empirical performance, and to compare our results with the **PASTA** algorithm [26].

We evaluate the empirical performance of our proposed confidence-based algorithms under four representative settings that differ in coverage and model specification conditions. Settings 1 and 2 examine performance under varying coverage levels in the standard linear MNL model, while Settings 3 and 4 study robustness under misspecified mixture-MNL environments.

**Setting 1: Full Coverage.** In this setting, both the item features $X$ and the true parameter $\theta^\star$ are independently drawn from spherical uniform distributions. The observed assortments are uniformly sampled from all $K$-sized subsets of $[N]$, ensuring that every item is well covered. The experimental parameters are $N = 30$, $d = K = 10$, and $W = 1$, and results are averaged over 10 repetitions.

| $n$ | LCB-MLE | LCB-MLE-v2 | PASTA |
|------|---------|------------|-----------|
| 100 | 0.0073 | 0.0087 | 0.0077 |
| 300 | 0.0056 | 0.0088 | 0.0048 |
| 500 | 0.0051 | 0.0082 | **0.0041** |
| 1000 | 0.0051 | 0.0069 | 0.0046 |

Table 4: Comparison of SubOpt Gap (smaller is better) under full coverage, $N = 30$, $d = K = 10$, $W = 1$, over 10 repetitions.

Under full coverage, **PASTA** slightly outperforms both LCB-MLE variants. Moreover, LCB-MLE-v2—with its larger lower-confidence penalty—shows higher conservativeness and thus lower empirical efficiency. This observation aligns with our theoretical insight: when all items are well explored, excessive conservativeness can lead to under-selection and higher SubOpt gaps.

**Setting 2: Partial Coverage.** We next consider a setting with partial coverage to examine robustness when the observation set does not fully span the item space. We use the same setup for $(X, \theta^\star)$, $N$, $K$, and $d$ as in Setting 1, and fix the total number of observations at $n = 500$. For each $n^\star$, assortments are constructed by including $n^\star$ items from the optimal set and filling the remaining $K - n^\star$ items with randomly sampled non-optimal ones. Increasing $n^\star$ therefore improves the effective coverage of the optimal set.

| $n^\star$ per $S$ | LCB-MLE | LCB-MLE-v2 | PASTA |
|------|---------|------------|-----------|
| 2 | 0.0050 | 0.0057 | 0.0047 |
| 4 | 0.0004 | 0.0004 | 0.0021 |
| 6 | 0.0007 | **0.0000** | 0.0022 |
| 8 | 0.0003 | **0.0000** | 0.0019 |

Table 5: Comparison of SubOpt Gap (smaller is better) under partial coverage.

Under partial coverage, both LCB-MLE variants consistently outperform **PASTA**. In particular, LCB-MLE-v2 achieves the smallest SubOpt Gap, suggesting that stronger conservativeness improves robustness when data coverage is limited. These results reveal a complementary pattern: LCB-MLE performs well in well-covered settings, whereas LCB-MLE-v2 is more effective in partially observed regimes.

**Setting 3: Misspecification under mixture of MNL I.** In this setting, we consider a mixture MNL model with two subgroups, where the second subgroup is selected with probability $\mu$, and the total sample size is fixed at $n = 500$. The generation of $(X, \theta^\star)$ and parameters $(N, K, d)$ follows

| $\mu$ | LCB-MLE | LCB-MLE-v2 | PASTA |
|------|---------|------------|-------|
| 0.05 | 0.9312 | 0.9280 | 0.9316 |
| 0.20 | 0.9279 | 0.9248 | 0.9277 |
| 0.50 | 0.9246 | 0.9234 | 0.9246 |
| 0.70 | 0.9263 | 0.9267 | 0.9262 |
| 0.90 | **0.9312** | **0.9312** | 0.9309 |

Table 6: Comparison of Expected Revenue (larger is better) under mixed MNL model I.

the same configuration as in Setting 1. Each assortment consists of 5 items sampled from the first subgroup's optimal set and 5 from the rest.

Since computing the true optimal assortment under a mixture MNL is intractable, we evaluate performance using the *expected revenue* instead of SubOpt Gap. As $\mu$ approaches $0.5$, model misspecification becomes more severe and the performance of all methods slightly degrades. Overall, all three algorithms perform comparably, showing that our confidence-based methods remain stable under mild model mismatch.

| $\delta$ | LCB-MLE | LCB-MLE-v2 | PASTA |
|------|---------|------------|-------|
| 0.1 | 0.9332 | 0.9338 | 0.9326 |
| 0.2 | 0.9328 | **0.9330** | 0.9322 |
| 0.5 | 0.9313 | 0.9316 | 0.9308 |
| 0.7 | 0.9311 | 0.9314 | 0.9307 |

Table 7: Comparison of Expected Revenue (larger is better) under mixed MNL model II.

**Setting 4: Misspecification under mixture of MNL II.** This experiment uses the same parameter setup as in Setting 3, with the mixing probability fixed at $\mu = 0.5$. The distance between the two subgroups' parameters $\theta^\star$ is controlled by a parameter $\delta$. As $\delta$ increases, model misspecification becomes more pronounced and overall performance declines. In this case, LCB-MLE-v2 slightly outperforms LCB-MLE and PASTA, indicating that stronger conservativeness may improve robustness under severe model mismatch.

