# OpenReview forum: "Improved Confidence Regions and Optimal Algorithms for Online and Offline Linear MNL Bandits"
_NeurIPS.cc/2025/Conference — NeurIPS 2025 poster_

### Official Review · Reviewer_NL9u · 2025-06-29

**Clarity:** 3
**Significance:** 2
**Originality:** 2
**Rating:** 4
**Confidence:** 4

**Summary:**

This paper addresses the assortment-optimization problem under a linear-utility multinomial logit (MNL) choice model. It develops algorithms for both the offline setting—where previously collected data are used to minimize the suboptimality gap—and the online setting—where sequential interactions are used to minimize cumulative regret. First, the authors establish an improved confidence region for the unknown true MNL parameter. In the offline setting, they propose a pessimistic algorithm based on per-item coverage and establish an improved suboptimality-gap bound that, unlike prior bounds, does not depend on the feature dimension $d$ or the problem-dependent constant $\kappa$ and does not require full coverage of the optimal assortment $S^*$. In the online setting, they design a SupCB-based algorithm that achieves cumulative regret $O(\sqrt{d T\log N})$.

**Questions:**

1. How does Assumption 1 differ from assuming $\set{(X_t,S_t)}$ are i.i.d.? Can you give an example where $(X_t,S_t)$ may be dependent over time but the $i_t$ remain conditionally independent?

2. In Line 172, what precisely is the improvement in the burn-in condition compared to [42]? Does satisfying [42]’s burn-in condition imply that the condition of this paper also holds?

Minor Comments

1. **Typo** (Line 65): “shows” → “show.”
2. **Typo** (Line 297): “a improved” → “an improved.”
3. **Missing Definition** (Line 167): $H^\lambda\_D(\theta^*)$
4. **Notation Inconsistency** (Line 203): Clarify $p_j(S^*)$ notation.

**Ethical Concerns:**

["NO or VERY MINOR ethics concerns only"]

**Final Justification:**

The author-reviewer discussion has satisfactorily addressed my questions, and I appreciate that the authors have appropriately conducted the additional experiments requested by other reviewers. However, I believe that improvements in the readability of the current manuscript are still needed. Therefore, I will maintain my current evaluation.

**Limitations:**

The authors acknowledge the computational inefficiency of the SupCB algorithm in Section 5.2, which may hinder real-world adoption.

**Paper Formatting Concerns:**

No major issues noted; the overall presentation is neat and professional.

**Quality:**

2

**Strengths And Weaknesses:**

[Quality]
- The theoretical proofs for the paper’s main contributions (Theorems 1, 2, and 3) are presented in detail in the appendix, and the necessary auxiliary lemmas are explained clearly. Due to time constraints, I did not rigorously verify every proof in the appendix; I focused mainly on the proof of Theorem 1 and did not find any obvious technical errors. Moreover, the advantages of the proposed methods over prior work in both the offline and online settings are well articulated, and the tightness of the algorithms is supported by the lower bound in the uniform‐revenue case (Theorem 4).

[Clarity]
- The main text is relatively well written and easy to understand, but it would be beneficial to include a proof sketch before each main‐theorem proof in the paper or appendix. I also recommend a thorough grammar check, as there are minor grammatical errors. Additionally, adding a Conclusion section to the main text would improve readability.


[Significance & Originality]
- This paper proposes improved statistical guarantees for offline and online settings based on an MLE‐derived confidence region for the true MNL choice parameter. While recent MNL‐bandit literature has explored $\kappa$‐improved regret bounds, most use online‐update methods or incur an extra $\sqrt{d}$ factor when using MLE. In contrast, this work establishes $\kappa$- and $d$-independent improvements under an MLE framework, representing a meaningful theoretical advance.

__Weaknesses__
- There is no experimental validation to support the theoretical results.
- Although the paper achieves $\kappa$-improved sub-optimality gaps and regret bounds in both settings, verifying the burn-in condition (e.g., whether Eq. (4) holds) requires prior knowledge of $\kappa$, which may not be available in practice. This represents a critical weakness compared to \[35, 44].
- As noted by the authors, SupCB‐type algorithms are computationally inefficient. Moreover, Algorithm 2’s regret bound depends on the total number of items $N$, limiting applicability when $N$ is very large (or infinite). Thus, although tight in $d$, the proposed algorithm may be impractical for large‐scale problems.

---

> ### Author Rebuttal · Authors · 2025-07-30
>
> Thank you for your valuable questions and suggestions! We greatly appreciate your feedback and provide our point-by-point responses below.
>
> **Q1: Difference between Assumption 1 and i.i.d. assumption.**
>
> Compared to the i.i.d. assumption over $(X_t,S_t)$, Assumption 1 requires only **independence among $(i\_t)\_{t=1}^T$ conditional on $\{ (X\_t,S\_t) \} \_{t=1}^T$**. This still allows complicate dependence among $(X\_t,S\_t)$ across $t$.
>
> An example of generating samples that satisfy conditional independence in $i_t$ is the SupCB algorithm we present in Section 4, originally inspired by [8]. The key to achieving this property is ensuring that the decision-making process does not depend on past choice observations. After the warm-up phase, each layer of Algorithm 2 uses only the $(X_t, S_t)$ information from samples collected during step (a) to compute the confidence region, which then guides future exploration actions in subsequent rounds of step (a). This design ensures that the samples used to construct the confidence region are independent of the observed choices at this layer, thereby satisfying the conditional-independence requirement on $(i_t)$.
>
>
> **Q2: Comparison of our burn-in condition to those in [42].**
>
> Under our notation with $\lambda = 0$, the confidence bound result in [42] can be stated as follows:
>
> Under the burn-in condition,
> $$ \lambda\_{\min}(V_t) \gtrsim \kappa^{-4}d^2 \text{ with }  V_t = \sum_{k=1}^t \sum_{j\in S_t} x_{k,j}x_{k,j}^\top, \tag{R1}$$
>
> it holds with probability at least $1-\delta$ that
>
> $$ \lvert x^\top(\hat\theta_D - \theta^\star) \rvert \lesssim \kappa^{-1} \lVert x \rVert_{V_t^{-1}} \tag{R2}.$$
>
> Now we compare the result in [42] with ours:
> - **Burn-in condition.** Comparing the burn-in condition in (R1) and the corresponding confidence bound in (R2) with our results, note that from the inequality
> $$ H_t(\theta^\star) \succeq \kappa V_t,  \tag{R3} $$
> we have
>
> $$\text{(R1) holds }\implies  \lambda_{\min}(H_t(\theta^\star)) \geq  \kappa  \lambda_{\min}(V_t) \gtrsim  \kappa^{-3} d \gtrsim d \implies  \lVert x \rVert_{H_t(\theta^\star)^{-1}} \lesssim 1/\sqrt{d}, \forall \lVert x \rVert_2\leq 1.  $$
>
> Therefore, the burn-in condition required in [42] holds implies that our burn-in condition is also satisfied.
>
> - **Confidence interval length.** By (R3), we have for any $x$,
>
> $$ \lVert x\rVert_{H_t^{-1} (\theta)}  \lesssim  \kappa^{-1} \lVert x \rVert_{V_t^{-1}} ,$$
>
> this shows our confidence interval length is smaller than (R2) result provided in [42].
>
> **Typos.**
>
> We thank the reviewer for pointing out the typos and limitations mentioned in the weaknesses and minor comments. We will correct the typos and further acknowledge the limitations regarding the $\kappa$ prior in the revised version. Specifically, we will discuss the related results from [35, 44] below our theoretical guarantee for the initialization phase in Algorithm 2 to improve clarity.
>
> **Empirical results**
>
> Regarding the experiment results, we have added several numerical experiments, including the
>
> - *Illustration of burn-in condition:* We provide an empirical illustration of our confidence bounds under varying burn-in constants, demonstrating the sensitivity to the burn-in condition in Theorem 1. Please refer to our response to Reviewer nFhi for detailed results and setup.
>
> - *Offline MNL bandits:* We compare our approach with the PASTA algorithm [26] for offline linear MNL bandits under full coverage, partial coverage, and misspecified settings. Details can be found in our response to Reviewer LNHW.
>
> We hope these additional experiments can offer empirical evidence to complement our theoretical contributions.

---

> > ### Author Response · Authors · 2025-08-04
> >
> > Dear Reviewer,
> >
> > To further clarify your Question 1, we would also provide an additional toy example on a 3 time-step sequence that may be helpful to understand:
> >
> > **A $t=3$ example on conditional independence.**
> >
> > Considering a three-step sampling procedure to obtain $(S\_1,i\_1)$, $(S\_2,i\_2)$ and $(S\_3,i\_3)$ with N = 5 and arbitrary $d,\theta^\star, [X\_i]\_{i\in [N]}$ :
> >
> > Step1: Sampling uniformly from [N] twice without replacement to obtain $S\_1$, and observe the feedback $i\_1$ generated from the MNL model.
> >
> > Step2: Sampling uniformly from $[N]\setminus S\_1$ twice without replacement to obtain $S\_2$, and observe the feedback $i\_2$ generated from the MNL model.
> >
> > Step3: Set $S_3$ as the remained item set not sampled by $S_1,S_2$.
> >
> > We can now discuss the dependence on $(S\_1,S\_2)$ and $(i\_1,i\_2)$:
> >
> > - **Fact 1: $S\_1, S\_2,S\_3$ are dependent:** We can easily check this fact by observing that $S\_1, S\_2 ,S_3$ can never have overlap.
> > - **Fact 2: $i\_1,i\_2$ are dependent:** We can also check this fact by observing that condition on $i\_1 \neq 0, i\_2$  will never equal to $i\_1$.
> > - **Fact3:** $i\_1,i\_2$ are **independent conditionally on $(S\_1,X\_1,S\_2,X\_2, S\_3)$:** To see this fact, noticing that
> >
> > $$ P(i\_1, i\_2 |S\_1,X\_1,S\_2,X\_2,S\_3 ) = P(i\_1, i\_2, S\_3 | S\_1,X\_1,S\_2,X\_2 ) / P(S\_3 | S\_1,X\_1,S\_2,X\_2).\tag{T1}$$
> >
> > Now by $S\_3$ only depend on $i\_1,i\_2$ through $S\_1,S\_2,$ we have
> >
> > $$P(i\_1, i\_2, S\_3 | S\_1,X\_1,S\_2,X\_2 ) = P (i\_1, i\_2|  S\_1,X\_1,S\_2,X\_2 )  P( S\_3 | S\_1,X\_1,S\_2,X\_2 ). \tag{T2}$$
> >
> > Then by (T1), we have
> >
> > $$ P(i\_1, i\_2 |S\_1,X\_1,S\_2,X\_2,S\_3 )  = P(i\_1, i\_2 | S\_1,X\_1,S\_2,X\_2 ) .$$
> >
> > Again, by $i\_2$ only depend on $i\_1$ through $S_2,$ we have
> >
> > $$ P(i\_1, i\_2 | S\_1,X\_1,S\_2,X\_2 )  =  P(i\_1 | S\_1,X\_1,S\_2,X\_2 )  P(i\_2 | S\_1,X\_1,S\_2,X\_2 ).  $$
> >
> > **This illustrates that $i_1$ and $i_2$ are conditionally independent, even if they are dependent unconditionally.** It is also worth noting that condition (T2) generally fails when $S_3$ depends directly on $i_1$ or $i_2$—for example, if $S_3 = [i_2]$. We hope this provides intuition for why choice-independent decisions are necessary in the SupCB algorithm.
> >
> >
> > We hope our response (this one and the earlier one) has addressed your concerns. If you have any further questions, we would be happy to discuss them with you. Please feel free to reach out at your earliest convenience so that we have sufficient time to make further discussion.
> >
> > Best,
> >
> > Authors

---

> > > ### Comment · Reviewer_NL9u · 2025-08-05
> > >
> > > Thank you to the authors for their thoughtful and detailed responses. It is very helpful in deepening my understanding of the paper. I have no further questions.

---

> > > > ### Author Response · Authors · 2025-08-05
> > > >
> > > > Thank you for your feedback! We will include the related discussions in the revised version to help explain Assumption 1 and the comparison to [42].

---

### Official Review · Reviewer_LNHW · 2025-06-29

**Clarity:** 3
**Significance:** 4
**Originality:** 4
**Rating:** 5
**Confidence:** 3

**Summary:**

This paper studies data-driven assortment optimization under the MNL model. The authors first propose sharper confidence regions for the MLE of the linear MNL model, which improves upon the best known result by far. Based on these confidence regions, they design (a) a pessimistic (lower confidence bound) algorithm for the offline setting, which requires only item-level coverage rather than full assortment coverage; (b) an online version using a modified SupCB framework. These contributions represent theoretical advancements in both offline and online assortment optimization under the linear MNL model.

**Questions:**

1) While Theorem 1 assumes a burn-in condition, Proposition 2 suggests it can be relaxed at the cost of an additional \sqrt{d} factor. Could you clarify how sensitive the algorithm performance is to the tradeoff of using/relaxing this condition?

2) You note the algorithm is computationally inefficient due to the enumeration of assortments. Could the proposed method be adapted to more practical heuristics while retaining theoretical guarantees?

3) If the linear MNL model is misspecified (e.g., user choice follows a nested logit or a mixture model), how would the proposed methods degrade? Are there robustness guarantees?

4) While the work is theoretical, are there plans to validate the findings on synthetic or real-world e-commerce datasets to demonstrate actual performance improvements and coverage requirements? Moreover, could the analysis framework extend beyond assortment optimization, e.g., to recommendation systems or pricing models with similar choice structures?

**Ethical Concerns:**

["NO or VERY MINOR ethics concerns only"]

**Final Justification:**

Good paper. I’ll retain my positive rating.

**Limitations:**

Yes, the authors discuss key limitations throughout, particularly the dependency on the burn-in condition for tight bounds and the computational inefficiency of the SupCB-based algorithm.

This is a pure theoretical paper so no potential negative societal impact of their work.

**Paper Formatting Concerns:**

NA.

**Quality:**

4

**Strengths And Weaknesses:**

Strength: (1) The theoretical results are solid, clearly stated, and well-justified. The paper is relatively well written and easy to follow, despite the heavy amount of math. (2) The work pushes the boundary of what is known in both online and offline linear MNL bandits. Its improvements over prior state-of-the-art in both theory and algorithm design. I believe this paper makes a valuable contribution to the field.


Weakness: (1) The analysis requires a “burn-in” condition for sharper bounds, though this is partially addressed with a workaround at the cost of a \sqrt{d} factor. Practical implications (e.g., computational scalability in real-world settings) are briefly mentioned but not deeply explored. (2) Due to its technical depth, some sections (especially in the online algorithm) are dense and could benefit from additional intuition or streamlining for a broader audience. But overall none of the weakness is significant; this is a clearly a good paper and makes novel contribution.

---

> ### Author Rebuttal · Authors · 2025-07-31
>
> Thank you for your valuable questions and suggestions! We greatly appreciate your feedback and provide our point-by-point responses below.
>
> **Q1: The sensitivity of algorithm performance under burn-in conditions and the trade-off.**
>
> Thank you for this insightful question. In the revised version, we have included several numerical studies examining the sensitivity of our confidence interval (CI). Below, we highlight some key observations related to this issue and refer the reviewer to our response to Reviewer nFhi for additional details. For clarity, we denote the burn-in factor as
> $$ \zeta:= \max\_{k\leq t, j \in S\_k} \lVert x\_{kj} \rVert\_{H^{-1}(\theta^\star)} $$
> for simplicity.
>
> - **Sensitivity of CI result in Theorem 1.** While there is no clear phase-transition threshold $\alpha^\star$ for the burn-in constant $\zeta$—i.e., no sharp point at which the CI fails for $\zeta > \alpha^\star$ and holds for $\zeta < \alpha^\star$—our results show that the CI ratio increases with $\zeta$. This trend may indicate a potential violation of the our CI guarantee in Theorem 1 in the large $\zeta$ regime.
>
> - **Stability of the CI bound in Proposition 2( from [44]).** The CI ratio of [44] remains stable as $\zeta$ increases. This aligns with the theoretical guarantee in Proposition 2, which does not require a burn-in condition.
>
> - **Better Trade-off between the two CI in algorithm design.** In practice, we may wish to adaptively select between the two confidence intervals to perform well in both small and large $\zeta$ regimes.
>
> One way to enable such adaptivity is by computing a data-driven upper bound on $\zeta$: First, note that $V_t$, defined in line 170 of our paper, is computable. From the dominance relation $H_t(\theta^\star) \succeq \kappa V_t$, we have
> $$ \zeta = \max\_{k\leq t, j \in S\_k} \lVert x\_{kj} \rVert\_{H_t^{-1}(\theta^\star)} \leq  \kappa^{-1/2} \max\_{k\leq t, j \in S\_k} \lVert x\_{kj} \rVert\_{V_t^{-1}}. $$
> Letting
>  $$\hat{\zeta}:= \max\_{k\leq t, j \in S\_k} \lVert x\_{kj} \rVert\_{V_t^{-1}},$$
> we arrive at a fully data-driven quantity. Then for any threshold $\alpha > 0$, a sufficient condition for $\zeta \leq \alpha$ is $\hat{\zeta} \leq \kappa^{1/2} \alpha$. Moreover, if an upper bound on the parameter norm $W$ is available, we can obtain a (possibly conservative) lower bound on $\kappa$, making this condition verifiable from observed data.
>
> In particular, we can simply switch to the CI from Theorem 1 if and only if $\hat{\zeta} \leq \kappa^{1/2} \alpha$, where $\alpha$ could be, for example, the burn-in threshold in Theorem 1. We hope this provides a practical guideline for balancing the two types of confidence bounds.
>
>
> **Q2: Practical heuristics to improve computational efficiency for Algorithm 2.**
>
> Thank you for this interesting question! While several heuristics have been proposed in the MNL setting to improve enumeration efficiency in MNL bandits (e.g., the DP-based approach and greedy swapping heuristics discussed in Section 5 of [18]), **we believe that developing similar heuristics in our setting may be significantly more challenging.**
>
> The main difficulty lies in the nature of the assortment space $\mathcal{A}\_\ell$: Unlike [18], where enumeration is performed over all $K$-sized assortments—a well-structured and easily verifiable set—the collection $\mathcal{A}\_\ell$ in our setting is exponentially large and *unstructured*. This means that for any given assortment $S$, determining whether it belongs to $\mathcal{A}_\ell$ may require searching through (or storing) all $\lvert \mathcal{A}\_\ell \rvert$ elements. In contrast, in [18], one can simply check whether $S$ is of size $K$ to verify feasibility. This lack of structure makes heuristic design for our setting particularly difficult.
>
> We will include the above discussion in the revised version to clarify the challenges involved.
>
>
> **Q3\& Q4(a): Empirical studies and performance under model misspecification.**
>
> We thank the reviewer for pointing out direction for taking numerical studies. While our algorithm 2 is computationally in-efficient thus hard to take empirical studies. We have added several numerical studies on our Algorithm 1 in the offline setting. We have compared Algorithm 1 with two types of CI( Theorem 1 and Proposition 2), denoted by LCB-MLE and LCB-MLE-v2, to illustrate the trade-off author raised in question 1. We have also compared our result with the PASTA algorithm in [26]. We have conduct the experiment in 4 different settings, as detailed below:
>
>
> **Setting 1: Full Coverage Setting.**
>
> **Table 1: Comparison of SubOpt Gap (smaller is better) under full coverage, $N = 30, d = K =10, W = 1$, over 10 repetitions.**
>
>
> | n | LCB-MLE | LCB-MLE-v2 | PASTA  |
> |---------|---------|------------|--------|
> | 100     | 0.0073  | 0.0087     | 0.0077 |
> | 300     | 0.0056  | 0.0088     | 0.0048 |
> | 500     | 0.0051  | 0.0082     | 0.0041 |
> | 1000    | 0.0051  | 0.0069     | 0.0046 |
>
>
> In this setting, both the item features $X$ and the true parameter $\theta^\star$ are independently drawn from spherical uniform distributions. The observed assortments are sampled uniformly from all possible $K$-sized subsets of $[N]$. Under this setup, every item is guaranteed to be sufficiently covered. We observe that PASTA outperforms both LCB-MLE algorithms, and that LCB-MLE-v2—with its larger LCB penalty—performs worse than LCB-MLE. A possible explanation is that the increased penalty in LCB-MLE-v2 results in overly conservative decisions in this well-covered setting.
>
>
>
>
> **Setting 2: Partial Coverage Setting.**
>
> **Table 2: Comparison of SubOpt Gap (smaller is better) under partial coverage**
>
> | $n^\star$  | LCB-MLE | LCB-MLE-v2 | PASTA  |
> |------|---------|------------|--------|
> | 2    | 0.0050  | 0.0057     | 0.0047 |
> | 4    | 0.0004  | 0.0004     | 0.0021 |
> | 6    | 0.0007  | 0.0000     | 0.0022 |
> | 8    | 0.0003  | 0.0000     | 0.0019 |
>
> In this setting, we use the same $X$ and $\theta^\star$ generation and $N,K,d$ as in Setting 1, with a fixed sample size of $n = 500$. For each $n^\star$, assortments are constructed by sampling $n^\star$ items from the optimal set and the remaining $K - n^\star$ from other items, creating partial coverage. As $n^\star$ increases, coverage of the optimal set improves. We observe that both LCB-MLE algorithms outperform PASTA, with LCB-MLE-v2 performing best, likely due to its stronger conservativeness under partial coverage.
>
>
> **Setting 3: Misspecification under mixture of MNL I.**
>
> **Table 3: Comparison of Expected Revenue (larger is better) under mixed MNL model I.**
> | $\mu$ | LCB-MLE | LCB-MLE-v2 | PASTA  |
> |-------------|---------|------------|--------|
> | 0.05        | 0.9312  | 0.9280     | 0.9316 |
> | 0.20        | 0.9279  | 0.9248     | 0.9277 |
> | 0.50        | 0.9246  | 0.9234     | 0.9246 |
> | 0.70        | 0.9263  | 0.9267     | 0.9262 |
> | 0.90        | 0.9312  | 0.9312     | 0.9309 |
>
> In this setting, we consider a mixture MNL model with two subgroups, where the second group is selected with probability $\mu$, and total sample size is $n = 500$. The generation of $X$, $\theta^\star$, and parameters $N$, $K$, $d$ follows Setting 1. Each assortment consists of 5 items sampled from the first group's optimal set and 5 from the rest.
>
> Since computing the optimal assortment under a mixture MNL is intractable, we evaluate performance using expected revenue instead of SubOpt.
>
> As $\mu$ approaches 0.5, model misspecification increases and performance degrades. All three algorithms perform similarly across settings.
>
> **Setting 4: Misspecification under mixture of MNL II.**
>
> **Table 4: Comparison of Expected Revenue (larger is better) under mixed MNL model II.**
> | Delta | LCB-MLE | LCB-MLE-v2 | PASTA  |
> |-------|---------|------------|--------|
> | 0.1   | 0.9332  | 0.9338     | 0.9326 |
> | 0.2   | 0.9328  | 0.9330     | 0.9322 |
> | 0.5   | 0.9313  | 0.9316     | 0.9308 |
> | 0.7   | 0.9311  | 0.9314     | 0.9307 |
>
> In this setting, we use the same parameter setup as in Setting 3, with $\mu$ fixed at $0.5$. The distance between the two subgroups' parameters $\theta^\star$ is controlled by a parameter $\delta$.
>
> As $\delta$ increases, model misspecification becomes more severe and performance degrades. In this setting, LCB-MLE-v2 outperforms LCB-MLE, which outperforms PASTA. This suggests that stronger conservativeness may improve performance under this type of misspecification.
>
> **Q4(b): Extensions the analysis framework to other settings with similar structure.**
>
> Yes, our results can be extended to other settings. One promising direction is the joint assortment and pricing framework considered in [R1, R2], where actions include both assortment selection and item pricing $p$. In this model, the final choice outcome can still be viewed as a sample from a linear MNL model incorporating both utility parameters and prices. This suggests that our Theorem 1, particularly the confidence bound construction, can also apply.
>
> However, this extension poses computational challenges, as optimizing over both assortments and prices increases the overall complexity. Combining this with UCB/LCB-based algorithm design, as discussed in [R1, R2], may introduce further complexity and warrants future investigation.
>
>
> [R1] Miao, Sentao, and Xiuli Chao. "Dynamic joint assortment and pricing optimization with demand learning." Manufacturing & Service Operations Management 23.2 (2021): 525-545.
>
> [R2] Erginbas, Yigit Efe, Thomas Courtade, and Kannan Ramchandran. "Online Assortment and Price Optimization Under Contextual Choice Models." The 28th International Conference on Artificial Intelligence and Statistics.

---

> > ### Comment · Reviewer_LNHW · 2025-08-06
> >
> > I appreciate the authors' clarification on my questions. They help me understand the paper better. As mentioned in my original review, "none of the weakness is significant; this is a clearly a good paper and makes novel contribution." Thus, I retain my positive score for the paper.

---

> > > ### Author Response · Authors · 2025-08-06
> > >
> > > Thank you for your feedback and valuable suggestions! We will include the related discussions in the revised version.

---

### Official Review · Reviewer_iMxr · 2025-07-02

**Clarity:** 3
**Significance:** 3
**Originality:** 3
**Rating:** 5
**Confidence:** 3

**Summary:**

The paper studies the assortment optimization problem: choosing a maximizing subset (of revenue) of size K among N arms.The authors provide a new, sharper, confidence bound for the multinomial logit based on a variance-aware analysis. The authors then use the improved confidence region of the algorithm to estimate the optimal subset using a pre-collected dataset $\mathcal{D}$. Then they propose an algorithm for an online variant of the problem: SupCB when the number of arms scales exponentially with d.

**Questions:**

There is theoretical motivation for this work but given the advances in other fields, how does this work practically compare with others? How well does this method do in recommendation on existing datasets?

**Ethical Concerns:**

["NO or VERY MINOR ethics concerns only"]

**Final Justification:**

The authors comments and the detailed discussion and responses have made me increase my score

**Limitations:**

Yes

**Quality:**

3

**Strengths And Weaknesses:**

While the work is theoretical, given the advances in adjacent fields, it would be good to do comparisons on existing algorithms and even benchmarks that aren't related to bandits. Suppose you are given a subset of featurized data points and you are asked to estimate the optimal subset, you could use a neural network based algorithm to rank the top K items. This will not have any guarantees like the user provides but how would it perform in practice compared to this algorithm? Given the motivation for the work it would help determine if this paper is of purely theoretical value: the improved, sharper confidence bound or if there is a more practical application.

Clarity: $\kappa$ is mentioned in the abstract and introduction without a formal definition. This makes the paper harder to read as one of the improvements is on the dependence on that parameter.

---

> ### Author Rebuttal · Authors · 2025-07-31
>
> Thank you for raising these important empirical concerns and suggestion on introducing formal definition of $\kappa$ in abstract and introduction!
>
> We will add a brief definition of $\kappa$ to improve clarity in the revised version.
>
> Regarding the practical applications, we acknowledge that our work has limitations in terms of direct applicability to real-world systems such as recommendation engines. A key assumption in our setting is the **presence of a known linear feature representation of items**, where the only unknown is the linear interaction parameter $\theta^\star$.
> This contrasts with the broader literature on recommendation systems, where representation learning plays a central role, and the optimal recommendations are often learned jointly with the representations—typically through end-to-end models such as deep neural networks, as the reviewer mentioned.
>
> This fundamental difference limits the direct applicability of our algorithm and theoretical results to the types of problems addressed in modern recommendation systems. In particular, the challenge of learning feature representations from data is central in those systems but falls beyond the scope of our work, which focuses on the linear MNL bandit setting with known features. We view integrating feature learning into this framework as an important and promising direction for future research, which could potentially start with neural network–based MNL bandit formulations, such as the recent work [R1]—but this is beyond the scope of the present study.
>
> Finally, although our emphasis is primarily theoretical, we have included several synthetic experiments that we hope will be of interest to reviewers. These include:
>
> - evaluating how the burn-in condition affects the confidence interval in Theorem 1, discussed in our response to reviewer nFhi;
>
> - comparing our method with the PASTA algorithm, a benchmark from prior work on offline linear MNL models, discussed in our response to reviewer LNHW;
>
> - assessing the robustness of our method under model misspecification, discussed in our response to reviewer LNHW.
>
> We hope these additional experiments can provide several empirical supports for our theoretical work.
>
> [R1] Bae, Seoungbin, and Dabeen Lee. "Neural Logistic Bandits." arXiv preprint arXiv:2505.02069 (2025).

---

> > ### Comment · Reviewer_iMxr · 2025-08-04
> > **Post-rebuttal update**
> >
> > Thanks for your clarification. Based on your comments and reading through the remaining discussion, I will update my review to reflect.

---

> > > ### Author Response · Authors · 2025-08-05
> > >
> > > Thank you for your feedback! We will acknowledge the practical limitations and include the related discussions in the revised version.

---

### Official Review · Reviewer_nFhi · 2025-07-03

**Clarity:** 2
**Significance:** 2
**Originality:** 2
**Rating:** 5
**Confidence:** 3

**Summary:**

In this paper, the authors consider the data-driven assortment optimization problem under the linear multinomial logit (MNL) choice model.
The first main result of the paper is the derivation of an improved confidence set associated with a regularized MLE of the parameter $\theta^*$. This improvement is achieved by keeping track of the behavior of the empirical Hessian instead of relying on the worst-case condition number parameter as in prior work. Using this CI, the authors first propose and analyze an LCB-based offline assortment optimization algorithm, and then develop a sample-splitting based online regret minimization algorithm.

**Questions:**

**Q1 Empirical comparison of the new CI:** You mentioned in Line 174 that there exist CIs that do not require the burn-in conditions but have to pay an extra $\sqrt{d}$ factor. Can you include some small-scale numerical experiments that compare your CI with that of [44, 3, 35] to see how they behave before and after the burn-in periods.

**Q2 Comparison with [31]:** At a glance, the general form of the confidence set derived in Theorem 1 looks similar to the one derived by Jun et al. [31] in their Theorem 1 for the case of $K=1$. Can you include some discussion on the main challenges in generalizing the ideas of [31] for any $K \geq 1$. This will help us better understand the technical contributions of the paper.

**Q3 Algorithmic details:** Perhaps I missed it in my reading, but how is the  "exploration length $n_0$" used in Algorithm 2? Also, the condition (4) seems to require the knowledge of the parameter $\kappa$. Is that something that is usually known? Also, in Line 293 it is mentioned that Algorithm 2 is "computationally inefficient": can you present the exact computational complexity of the algorithm?

Minor Questions/Comments

* In the abstract the abbreviations MNL and MLE are used without introduction. Can you fix that?
* Line 3: "that removing the explicit": Should it be "that removes the explicit"?
* Line 8: "where $n_{S^\*}$ the number of times": Should it be "where $n_{S^*}$ denotes the number of times"?
* Line 162: What is $i$ in the displayed equation? Should it be $\log p_j$ instead of $\log p_i$?

**Ethical Concerns:**

["NO or VERY MINOR ethics concerns only"]

**Final Justification:**

I am satisfied with the response of the authors to my questions, and I now have a better understanding of the technical contributions (w.r.t. [31] for example). Overall, my impression of the paper is positive, and I will retain my score as I think this is a nice incremental contribution to the literature.

**Limitations:**

Yes.

**Paper Formatting Concerns:**

None.

**Quality:**

3

**Strengths And Weaknesses:**

Strengths

* The derivation of a more refined concentration result for the MLE that does not depend on the worst-case condition parameter $\kappa$ is an interesting contribution.

* By leveraging the improved concentration result of Theorem 1, the authors obtain tighter suboptimality gap (Theorem 2) bound in the offline setting, and an improved regret bound (Theorem 3) in the online setting.

Weaknesses

* While I did not observe any significant weaknesses, I think the results in this paper may not be immediately useful from a practical perspective. This is because they require a potentially large burn-in period to ensure Hessian conditioning, and furthermore, both the offline and online algorithms may become computationally infeasible in larger-scale problem instances.

---

> ### Author Rebuttal · Authors · 2025-07-30
>
> Thank you for your valuable questions and suggestions! We greatly appreciate your feedback and provide our point-by-point responses below.
>
>
> **Q1. Empirical comparison of CI results.**
>
> When comparing our confidence interval results with those in [44, 3, 35], our bounds are consistently tighter in the sense that the confidence radius is smaller, due to a $\sqrt{d}$-factor improvement. However, this comes at the potential cost of requiring an additional burn-in condition and a conditional independence assumption.
>
>
> In the following simulation, we compare our confidence bounds with those in [44] under varying burn-in constants to illustrate the potential failure of our bounds when the burn-in condition is violated. We hope this illustration can help understand the underlying trade-off between CI tightness and risk of violation. Below, we first present the results and discussion, followed by the experimental setup.
>
> - **Result and Comments:**
>
> In the table below, we compare the CI ratio, defined as
>
> $$
> \text{CI ratio}:= \max\_i  \frac{\lvert \hat\theta_i  - \theta^\star \rvert }{\text{CI at }e_i}
> $$
> for confidence interval results derived in our Theorem 1 and [44].
> The CI ratio measures how tightly the theoretical confidence bound captures the actual estimation error. A smaller CI ratio indicates a lower likelihood of violating the theoretical CI guarantee.
>
>
> We make the test under different burn-in constants:
> $$
> \zeta := \max\_{k\leq n,j \in S\_k} \lVert x_{kj} \rVert\_{H^{-1}(\theta^\star) }.
> $$
>
> Theoretically, our CI result in Theorem 1 requires  $\zeta \lesssim 1/\sqrt{d}$ to holds and the CI result in [44] have no restriction on $\zeta$.
>
> |     value of  $\zeta$           | 0.130   | 0.134   | 0.141   | 0.160   | 0.335   |
> |--------------------|---------|---------|---------|---------|---------|
> | CI ratio in Theorem 1 | 2.563   | 2.831   | 3.147   | 4.052   | 5.617   |
> | CI ratio in [44]   | 3.495   | 3.480   | 3.420   | 3.270   | 2.852   |
>
> We observe that as $\zeta$ increases, our CI ratio in Theorem 1 also increases, indicating a higher likelihood of violating the theoretical bound. In contrast, the CI ratio in [44] remains stable as $\zeta$ grows, which aligns with the differing theoretical requirements of the two confidence bounds.
>
> - **Experiment Setup - Feature Set and Parameter:** For a given $d,K$(W.L.O.G. suppose $d$ is even) and $\beta >0$,  we construct the item-associated feature map set with $N = d$  as the following:
>
> - *Type-1 Items:* The first d-1 features $x_i$ for $i\leq d-1$ are given by the canonical basis $x\_i = e\_{i}$.
>
> - *Type-2 Items* The last feature $x\_{d}$ is given by $x\_{i} = (1,1,...,1)/\sqrt{d}$.
>
> - *Underlying Parameter $\theta^\star$* Given radius $W > 0,$ the $\theta^\star $ is set as $\theta^\star = (W,W,...,W)/\sqrt{d}.$
>
> - **Experiment Setup - Assortment Sets:** With above construction of feature sets, we construct two types of assortments as the following:
>
> - *Type-1 Assortment:* A single-item assortment containing only the type-2 item $S_0 = \{ d \}$ as a single-item set that only contains item $d$.
> - *Type-2 Assortments:* Denote $m = \lceil d/K \rceil$, then for every $1\leq k\leq m,$ we set the $k$-th type-2 assortment as
> $$ S\_k:= [ (m-1)K + 1, (m-1)K + 2, ..., \min ( mK, d )  ] $$
>
> - *Observed assortment sets.* With above types of assortment sets. Given any number $n_1,n_2$, the observed assortment sets(and corresponding features and choices generated by linear MNL model with $X,\theta^\star$ above) is set as $n_1$ copies of $\{S_1,..., S_m\}$  and $n_2$ copies of $S_0.$
>
> With above construction, we can fix the value of $n_1, n_2$ and moving $W$ to generate different environments with different level of burn-in constants $\zeta$. In particular, under our construction $\zeta$ is increasing as $W$ increases, as we shown in the table below:
>
> | Value of $W$ | 0.5   | 0.841 | 1.41  | 2.37  | 4     |
> |--------------------------|-------|-------|-------|-------|-------|
> | Value of $\zeta$         | 0.130 | 0.134 | 0.141 | 0.160 | 0.335 |
>
>
> This explains how we obtain environments to generate the results presented in table 1. Specfically, we select $d = 6, K = 3$ and $n1 = n2 =  500$ in the above experiment.
>
>
> **Q2. Technical comparison with [31].**
>
>
> Thank you for raising this suggestion and question! We would acknowledge that our analysis follows the same overall framework in generalizing Theorem 1 of [31]. However, the detailed calculations require new arguments that explore the geometric landscape and self-concordant properties of $K$-MNL losses, which is a central topic in recent theoretical studies on MNL bandits.
>
> More precisely, our main technical contribution lies in Lemma 6, where we carefully explore the curvature of the MNL loss to derive a confidence region that avoids dependencies on $K$ and $\kappa$. This improves previous arguments in [42]( the only prior work that considers $d$-free confidence bounds under the independence condition), and parallels the refinement technique in [36], which is developed for different purpose( to obtain sharp variance-dependent bounds).
>
>
> **Q3.  Algorithmic details**
>
> -**Initial length $n_0$ in Algorithm 2** Thank you for pointing out this issue! It should not be an input to Algorithm 2, and we will remove it in the revised version. We apologize for the confusion caused.
>
> -**Prior knowledge of $\kappa$** In general, we cannot assume a strong prior on $\kappa$. However, it is relatively reasonable to assume prior knowledge of an upper bound $W$ on the parameter norm, as is common in previous work. Given this, we can derive a conservative lower bound on $\kappa$ that scales as $\exp(-K W)$, which is sufficient to serve as an algorithm input.
>
> -**Computation Complexity.** The elimination step of the algorithm 2 requires to search over the whole $\mathcal{A}_l$, whose complexity scales as $O(N^K)$ in the worst case.
>
> **Q4. Writing Suggestions and Typos.**
>
> Thank you so much for the suggestion and pointing out those typos! You are absolutely right on the problems in line 3, 8, 162, we will correct those problems in the revised version. We will also clarify the terms 'MLE', 'MNL' when they first appears to improve clarity.

---

> > ### Comment · Reviewer_nFhi · 2025-08-05
> > **Response to authors**
> >
> > Thank you for your detailed response to my questions. It would be nice if you could include the empirical results, and a paragraph comparing your proof techniques with [31] in the final version.

---

> > > ### Author Response · Authors · 2025-08-05
> > >
> > > Thank you for your suggestion and feedback! We will add the related discussion and empirical results to the revised version.

---

### Decision · Program_Chairs · 2025-09-17

**Decision:**

Accept (poster)

**Comment:**

This paper improves the performance bound of Linear MNL Bandits.

The contribution is very clear and solid. The improved concentration result is very relevant and a meaningful progress.